# INSERTION LANGUAGE MODELS: SEQUENCE GENERATION WITH ARBITRARY-POSITION INSERTIONS

## ABSTRACT

Autoregressive models (ARMs), which generate sequences by predicting tokens from left to right, have achieved significant success across a wide range of sequence generation tasks. However, they struggle to accurately represent sequences that require satisfying sophisticated constraints or whose sequential dependencies are better addressed by out-of-order generation. Masked Diffusion Models (MDMs) address some of these limitations, but MDMs struggle to generate variable length sequences and cannot handle arbitrary infilling constraints when the number of tokens to be filled in is not known in advance. We revisit the idea of generation by insertion and introduce Insertion Language Models (ILMs), which learn to insert tokens at arbitrary positions in a sequence—that is, they select jointly both the position and the vocabulary element to be inserted. The ability to generate sequences in arbitrary order allows ILMs to accurately model sequences where token dependencies do not follow a left-to-right sequential structure, while maintaining the ability to infill and generate up to a variable length. To train ILMs, we propose a tailored network parameterization with a single transformer encoder and use a simple denoising loss. Through empirical valuation on planning tasks we demonstrate the aforementioned failure modes of ARMs and MDMs, and show that ILMs overcome these. Furthermore, we show that ILMs perform on par with ARMs and better than MDMs in unconditional text generation while offering greater flexibility than MDMs in arbitrary-length text infilling.

## 1 INTRODUCTION

Autoregressive models (ARMs), which predict subsequent tokens one-by-one in a "left-to-right" fashion, have achieved significant success in modeling natural language (Brown et al., 2020; Grattafiori et al., 2024). Their simplicity makes them easy to train and has enabled a rapid increase in model sizes (Kaplan et al., 2020). However, ARMs have several fundamental limitations. For example, they have fallen short on tasks that require complex reasoning and long-horizon planning (Bubeck et al., 2023; Valmeekam et al., 2024; Dziri et al., 2023), and they struggle to accurately model sequences that require satisfying sophisticated constraints (Sun et al., 2023). Recently, Masked Diffusion Models (MDMs) have been shown to overcome some of the limitations of ARMs (Ye et al., 2025; Sahoo et al., 2024; Lou et al., 2024; Nie et al., 2024; 2025). Although MDMs address some of the limitations of ARMs, departing from strictly left-to-right generation introduces new challenges. First, using the vanilla sampling algorithm (Sahoo et al., 2024) leads to unmasking multiple tokens simultaneously during generation which can violate token dependencies. For example, in the sentence "The chef added <mask> to the dessert to make it <mask>." if both the <mask> tokens are filled simultaneously, it can lead to a sentence that does not make sense, for example, "The chef added *sugar* to the dessert to make it *healthier*." However, if the tokens are filled sequentially, more appropriate sentences are generated, for example, "The chef added *sugar* to the dessert to make it *sweeter*." or "The chef added *berries* to the dessert to make it *healthier*." One may achieve sequential generation from MDMs by greedily unmasking the most confident position, but this leads to slow generation as production of a single token requires a full forward pass. Second, reliance on the number of masked tokens in the input reduces a model's usefulness when performing arbitrary infilling. For example, when presented with the sentence "The conference, <mask> was postponed." the model cannot generate "The conference, *originally planned for March*, was postponed." as the input has only one mask.

To overcome these limitations, we revisit the idea of insertion based sequence generation (Stern et al., 2019; Ruis et al., 2020) in the context of general language modeling, and introduce Insertion Language Models (ILMs), which use a simple denoising objective that involves dropping some tokens from the input sequence and learning to predict the missing tokens sequentially, one at a time. Unfortunately, estimates of the naive infilling denoising objective can have extremely high variance, which in turn can make training infeasible. To address this issue and allow efficient training, we introduce an approximate denoising training objective and a tailored parameterization of the denoising network. The key difference between ILMs and MDMs is that in ILMs, the dropped tokens are completely removed from the input sequence and are generated one at a time in reverse, whereas in MDMs, the dropped tokens are replaced by a $<$mask$>$ token.

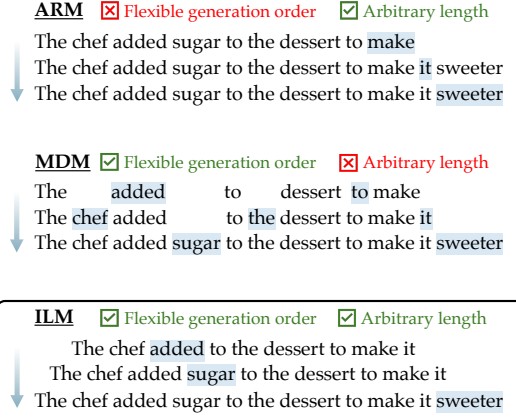

Figure 1: ARMs (top) generate variable-length sequences in a fixed left-to-right order. MDMs (middle) can add tokens in arbitrary order but require a fixed number of tokens to be masked. ILMs (bottom) generate sequences of arbitrary lengths in arbitrary order by inserting tokens.

Using a suite of carefully chosen synthetic tasks, we first demonstrate the failure modes of ARMs and MDMs, and show that ILMs overcome these. Specifically, in the task of path generation on star graphs (Bachmann & Nagarajan, 2024), ILMs can consistently generate the correct path even when ARMs and MDMs struggle—especially when the paths have variable length. We also find that ILMs outperform ARMs and MDMs on the difficult constraint satisfaction task of solving Zebra Puzzles (Shah et al., 2024). We also demonstrate the usefulness of ILMs for text generation and infilling. On medium-sized text corpora such as LM1B and TinyStories, we find that ILMs perform slightly better than MDMs on unconditional text generation task (measured using generative perplexity under Llama, and Prometheus LLM judge) and are competitive with ARMs. We also demonstrate the effectiveness of ILMs on infilling arbitrary length sequences on the same datasets.

To summarize, our main contributions are as follows:

1. We introduce Insertion Language Models (ILMs), which learn to insert tokens at arbitrary positions in a sequence and are able to handle strong dependencies between tokens.

2. We present a neural network parameterization and a simple denoising objective that enable the training of ILMs.

3. We conduct an empirical evaluation of the proposed method and find that ILMs outperform autoregressive and masked diffusion models on common planning tasks and are competitive with ARMs and MDMs on text generation tasks while offering greater flexibility on arbitrary-length text infilling compared to MDMs.

## 2 PRELIMINARIES

**Notation.** Capital letters are used to denote random variables (e.g. $\boldsymbol{X}$) and the corresponding lowercase letters are used to denote their values (e.g. $\boldsymbol{x}$). Boldface is reserved for non-scalars (vectors, matrices, etc.). Double square brackets are used to denote the set of natural numbers up to a specific number, that is, $[\![n]\!] = \{1, 2, \ldots, n\}$. The components of a non-scalar quantity are denoted using superscripts and subscript time index of a stochastic processes whenever applicable.

### 2.1 MASKED DIFFUSION MODELS

Let $\mathbb{V}$ denote the token vocabulary, a finite set, and $p_{\text{data}}$ be probability mass function on the set of sequences $\mathbb{V}^L$. Assume that there is an arbitrary and fixed ordering on set $\mathbb{V}$, using which we can use $\boldsymbol{e}_x$ to denote the indicator vector that is one at the index of token $x$ and zero otherwise. Furthermore, assume that the set $\mathbb{V}$ contains a special token, whose probability under $p_{\text{data}}$ is 0, called the *mask*

token denoted as $m$. The training objective for MDMs (Shi et al., 2024; Sahoo et al., 2024) can be written as the data expectation (i.e., $\boldsymbol{x}_0 \sim p_{\text{data}}$) of the following loss:

$$\mathcal{L}_\theta(\boldsymbol{x}_0) = \mathop{\mathbb{E}}_{\boldsymbol{x}_t \sim q_{t|0}(\cdot|\boldsymbol{x}_0)} \left[ \int_0^1 \frac{\alpha_t'}{1 - \alpha_t} \sum_{i=1}^L \delta(x_t^i, m) \log[\mu_\theta^{\text{mdm}}(\boldsymbol{x}_t, t)]_{x_0^i}^i \, dt \right],$$

where

$$q_{t|0}(\boldsymbol{x}_t \mid \boldsymbol{x}_0) = \prod_{i=1}^L \text{Cat}\left( \alpha_t \boldsymbol{e}_{x_0^i} + (1 - \alpha_t)\boldsymbol{e}_m \right) \tag{1}$$

is the transition probability of the noising process, and $\mu_\theta^{\text{mdm}} : \mathbb{V}^L \times [0,1] \to (\Delta^{|\mathbb{V}|-1})^L$ is the learned parametric denoiser that takes in the current noisy sequence and produces a categorical probability distribution over the vocabulary at each sequence position. Here $\Delta^{|\mathbb{V}|-1}$ denotes a categorical probability distribution over $\mathbb{V}$, and $[\mu_\theta^{\text{mdm}}(\boldsymbol{x}_t, t)]_j^i$ denotes the probability of $j$-th token from the vocabulary at $i$-th sequence position. Typically, the noising function $\alpha_t$ is a monotonically decreasing function defined on the interval $[0,1]$ with $\alpha_0 = 1$ (no noise) and $\alpha_1 = 0$ (most noise).

**Limitations of MDMs.** During inference, at time step $t$, with step size $s - t$, a subset of tokens is unmasked uniformly at random with probability $P(i) \propto \frac{\alpha_s - \alpha_t}{1 - \alpha_t} \delta(x_t^i, m)$, with their values sampled from $x_t^i \sim [\mu_\theta^{\text{mdm}}(\boldsymbol{x}_t, t)]^i$. This inference procedure has two shortcomings:

1. When the step size $s - t$ is large, many tokens are unmasked simultaneously, which could result in incoherent outputs due to violation of sequential dependencies .
2. Since the number of masks between any two unmasked tokens is fixed, the inference has no flexibility in terms of infilling length.

In the next section, we describe our proposed Insertion Language Model (ILM) that tries to address the limitations mentioned above.

## 3 INSERTION LANGUAGE MODEL

ILM generates sequences of arbitrary lengths in arbitrary order by inserting tokens, one-at-a-time, that is, at each generation step, it predicts an output token along with a position in the existing sequence where the new token is to be inserted. The model can also decide to stop at any step, deeming the sequence to be complete. ILM's ability to predict the insertion position obviates the need for placeholder mask tokens, and thus avoids the rigid fixed-length constraint imposed by the MDMs. Moreover, this also allows the model to pick the positions for generation in any order escaping the pitfalls of left-to-right generation as in ARMs. Figure 1 depicts the key difference between ILMs, MDMs and ARMs using example generation trajectories.

An ILM can be viewed as a denoising model whose noising process drops tokens as opposed to replacing them with mask tokens. Training such a denoiser requires marginalization over possible trajectories leading to the original sequence, which can be done using the Monte Carlo sampling and learning to reverse a single step of the noising process. However, that introduces high variance in the loss estimates (see Appendix D for more details). To avoid this issue, we use a biased training objective that makes direct use of all the dropped tokens in the original sequence in a single gradient step. Specifically, for a position between any two tokens in the partially predicted sequence, instead of estimating the token probabilities by marginalizing over all generation trajectories, we train the model to predict the normalized counts of each vocabulary item appearing between any two tokens, in the original sequence.

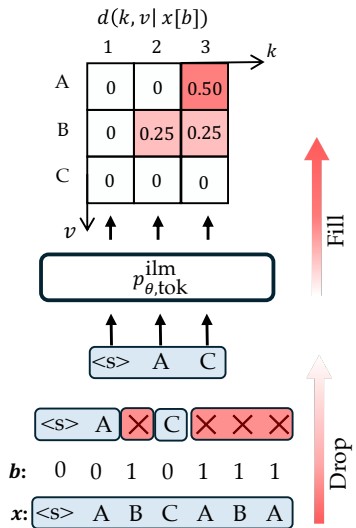

Figure 2: **ILM Training.** $\boldsymbol{x}$ is a training sequence, $\boldsymbol{x}[\boldsymbol{b}]$ is a subsequence obtained after dropping tokens. $d$ is the target insertion distribution, computed by counting the number of times each token appears in $\boldsymbol{x}$ between the $i_k$-th and $i_{k+1}$-th positions.

Our training objective is a sum of two components that are optimized simultaneously. First, the token insertion component $\mathcal{L}_{\text{tok}}^{\text{ilm}}(\theta; \boldsymbol{x})$. Second, a binary decision component $\mathcal{L}_{\text{stop}}^{\text{ilm}}(\theta; \boldsymbol{x})$, that decides when to stop generation and in turn governs the length of the sequence. Formally, let $\mathbb{B}_{L,n}$ be the set of bit vectors of length $L$ with exactly $n$ ones, and let $\boldsymbol{x}[\boldsymbol{b}]$ be the sequence obtained after *removing* the tokens corresponding to the ones in $\boldsymbol{b}$ from $\boldsymbol{x}$ (c.f. Figure 2 bottom). Let $p_{\theta,\text{tok}}(k, v \mid \boldsymbol{x}[\boldsymbol{b}])$ be the learned insertion probability of inserting token $v$ between positions $k$ and $k+1$, which is learned using

---

**Algorithm 1** ILM training

**Require:** Input example $\boldsymbol{x}$ of length $L$
1: Sample $n \sim U[\![L]\!]$
2: Sample $\boldsymbol{b} \sim q_{n|L}$
3: **Compute** $d(k, v; \boldsymbol{x}, \boldsymbol{b})$
4: $\mathcal{L}(\theta; \boldsymbol{x}) \leftarrow \mathcal{L}_{\text{tok}}(\theta; \boldsymbol{x}) + \mathcal{L}_{\text{stop}}(\theta; \boldsymbol{x})$
5: Update $\theta$ using gradient descent

---

$$\mathcal{L}_{\text{tok}}^{\text{ilm}}(\theta; \boldsymbol{x}) = - \underset{n \sim U[\![L]\!]}{\mathbb{E}} \ \underset{\boldsymbol{b} \sim q_{n|L}}{\mathbb{E}} \left[ \frac{1}{n} \sum_{k \in [\![L-n]\!]} c_{i_k, i_{k+1}}(v; \boldsymbol{x}) \log p_{\theta,\text{tok}}^{\text{ilm}}(k, v \mid \boldsymbol{x}[\boldsymbol{b}]) \right], \quad (2)$$

where $i_1, ..., i_{L-n}$ are the indices in $\boldsymbol{x}$ of the visible tokens after dropping tokens according to $\boldsymbol{b}$, $U[\![L]\!]$ is the uniform distribution over $\{1, ..., L\}$, $q_{n|L}(\boldsymbol{b}) = 1/\binom{L}{n}$ is the probability of selecting a bit vector of length $L$ with $n$ ones, and $c_{i_k, i_{k+1}}(v; \boldsymbol{x}) = \sum_{j=i_k}^{i_{k+1}-1} \delta(\boldsymbol{x}^j, v)$ is the number of times token $v$ appears in $\boldsymbol{x}$ between the $i_k$-th and $i_{k+1}$-th positions. Note that $d(k, v; \boldsymbol{x}, \boldsymbol{b}) := c_{i_k, i_{k+1}}(v; \boldsymbol{x})/n$ (with $n$ being the total number of tokens dropped), when summed over $k$ and $v$ gives 1. Therefore, we call it the target insertion distribution, which is usually quite sparse.

The second loss component is for learning a binary classifier $p_{\theta,\text{stop}}(S \mid \boldsymbol{x}[\boldsymbol{b}])$, where $S$ is binary random variable, which takes a partially noised sequence of tokens and predicts whether the sequence is complete ($S = 1$) or not.

$$\mathcal{L}_{\text{stop}}^{\text{ilm}}(\theta; \boldsymbol{x}) = - \underset{n \sim U[\![L]\!]}{\mathbb{E}} \ \underset{\boldsymbol{b} \sim q_{n|L}}{\mathbb{E}} \left[ \delta(\boldsymbol{b}, \boldsymbol{0}) \log p_{\theta,\text{stop}}^{\text{ilm}}(1 \mid \boldsymbol{x}[\boldsymbol{b}]) + (1 - \delta(\boldsymbol{b}, \boldsymbol{0})) \log p_{\theta,\text{stop}}^{\text{ilm}}(0 \mid \boldsymbol{x}[\boldsymbol{b}]) \right],$$

where $\boldsymbol{0}$ is the vector of all zeros. The overall training loss is the sum of the token insertion loss and the stopping loss. The stopping classifier and the denoiser share the transformer backbone and are trained simultaneously (see Section 3.1 for more details). The overall training procedure for ILM, shown in Algorithm 1, resembles that of MDMs, one extra step of computing the target insertion distribution (highlighted in bold).[1]

During inference, ILM inserts one token at a time as shown in Algorithm 2.[2] For step 4 in the algorithm, we can sample from the joint, or perform two-step sampling $k' \sim p_\theta^{ilm}(k \mid \boldsymbol{x}[\boldsymbol{b}])$ followed by $v' \sim p_\theta^{ilm}(v \mid \boldsymbol{x}[\boldsymbol{b}], k')$, where the latter approach allows us to use either top-k sampling or nucleus sampling (Holtzman et al., 2020) for each step separately.

---

**Algorithm 2** One step of ILM prediction

**Require:** Current sequence $\boldsymbol{x} = (\boldsymbol{v}, \boldsymbol{u})$, where $\boldsymbol{v}$ is the out-of-order sequence of tokens, and $\boldsymbol{u}$ is their corresponding real positions relative to one another, stopping threshold $\tau$
1: **if** $p_{\theta,\text{stop}}^{ilm}(1 \mid \boldsymbol{x}) > \tau$ **then**
2:     **return** $\boldsymbol{x}$
3: **end if**
4: $k', v' \sim p_{\theta,\text{tok}}^{ilm}(\cdot \mid \boldsymbol{x})$
5: $\boldsymbol{v}' \leftarrow \texttt{concat}(\boldsymbol{v}, v')$
6: **for** $i = 1$ to $\texttt{len}(\boldsymbol{u})$ **do**
7:     **if** $\boldsymbol{u}[i] > k'$ **then**
8:         $\boldsymbol{u}[i] \leftarrow \boldsymbol{u}[i] + 1$
9:     **end if**
10: **end for**
11: $\boldsymbol{u}' \leftarrow \texttt{concat}(\boldsymbol{u}, k' + 1)$
12: **return** $\boldsymbol{x}' = (\boldsymbol{v}', \boldsymbol{u}')$

---

### 3.1 PARAMETERIZATION

We parameterize $p_\theta$ using insertion logits computed using a standard transformer as follows Let $f_\theta^{\text{dec}} : \mathbb{V}^n \to \mathbb{R}^{n \times d}$ denote a transformer backbone, that is, a stack of transformer layers but without the final unembedding/linear layer. For each position $i \in [n]$ the corresponding output of the transformer backbone $f_\theta^{\text{dec}}(\boldsymbol{x})_i \in \mathbb{R}^d$ is passed through the unembedding layer $f_\theta^{\text{ins}} : \mathbb{R}^d \to \mathbb{R}^{|\mathbb{V}|}$ to get the insertion logits for each position in the sequence. In other words

$$s_\theta(k, v \mid \boldsymbol{x}[\boldsymbol{b}]) = f_\theta^{\text{ins}} \left( f_\theta^{\text{dec}}(\boldsymbol{x}[\boldsymbol{b}])_k \right)_v, \quad (3)$$

---

[1]Unlike in MDM training, where the mask is usually sampled on the GPU, we sample $\boldsymbol{b}$ and compute $d$ in the data pipeline on the CPU.

[2]The procedure can be implemented using tensor operations that can be performed on mini-batches.

which represents the unnormalized log probability (logit) for inserting token $v$ between $k$ and $k+1$ positions in the sequence $\boldsymbol{x}[\boldsymbol{b}]$. Finally, the join distribution over all possible insertions is given by

$$p_\theta^{\text{ilm}}(i_k, v \mid \boldsymbol{x}[\boldsymbol{b}]) = \frac{\exp(s_\theta(i_k, v \mid \boldsymbol{x}[\boldsymbol{b}]))}{\sum_{k=1}^{L-n} \sum_{v' \in \mathbb{V}} \exp(s_\theta(k, v' \mid \boldsymbol{x}[\boldsymbol{b}]))}. \tag{4}$$

The stopping probability is predicted using the output from a special `<stp>` that is always placed at the beginning of the input sequence. Therefore the input shown in Figure 2 looks like $\boldsymbol{x}[\boldsymbol{b}]$ =`<stp>` `` A C.

## 4 RELATED WORK

Diffusion models (Sohl-Dickstein et al., 2015; Ho et al., 2020; Song & Ermon, 2019) have emerged as a powerful alternative to ARMs for sequence generation tasks that require planning and need to follow constraints. Masked Diffusion Models (MDMs) have been shown to scale competitively to ARMs while addressing some of its key shortcomings (Austin et al., 2021; Campbell et al., 2022; Lou et al., 2024; Sahoo et al., 2024; Shi et al., 2024). However, as discussed in Section 2, due to the use of fixed length mask tokens, and simultaneous unmasking, these models, without additional inference time tricks, tend to generate incoherent sequences. To address this, Gong et al. (2024) propose to use a greedy strategy to select the tokens to unmask, Zheng et al. (2024) generalizes it to top-k sampling strategy, while Campbell et al. (2024) utilizes a flow-based formulation to introduce helpful stochasticity on top of the greedy sampling process.

All these approaches, rely on inference time techniques to elicit better samples. Ye et al. (2025) modify the MDM training objective by introducing an adaptive token-wise weight that helps the model identify the critical parts of the sequence. This objective, however, is only shown to work for synthetic tasks. Departing from this line of work, we propose a new parameterization and training objective. The MDMs are closely related to order-agnostic sequence models (Yang et al., 2020; Hoogeboom et al., 2021). The key difference between MDMs and order-agnostic models is that unlike MDMs, which can denoise the entire sequence in one go, order-agnostic models only generate one token at a time in an arbitrary order. Our model also generates the sequence by inserting tokens at arbitrary positions but is allowed to pick the position to insert the token.

The ability to insert tokens allow ILMs to perform infilling more naturally compared to ARMs. There has been only a handful of works that focus on the task of arbitrary length infilling using ARMs, most of which require specialized fine-tuning. Bavarian et al. (2022) introduces fill-in-the-middle training objective where ARMs are trained to take `<prefix><suffix>` as the left-context and is required to generate the `<middle>` part such that `<prefix><middle><suffix>` is a meaningful natural language sequence. While this approach enjoys the benefit of adapting an existing pre-trained ARM, its applicability is quite limited because the model is not capable of performing arbitrary infilling, for example, filling two blanks at separate places in the sequence. Gong et al. (2024) also proposes a method to adapt pre-trained ARMs to masked denoising models. However, once adapted, the model has the same limitations as MDMs. Please refer to Appendix A for an extended discussion.

## 5 EMPIRICAL EVALUATION

To highlight the key differences between ILMs, MDMs and ARMs, we consider two planning tasks: a generalized version of the synthetic planning task on star shape graphs introduced in Bachmann & Nagarajan (2024) and Zebra Puzzles (Shah et al., 2024). To demonstrate the effectiveness of ILM beyond synthetic planning task, we also perform unconditional text generation and infilling, for which we train the model on two language modeling datasets with different characteristics: (1) The One Billion Word Benchmark (LM1B) and (2) TinyStories (Eldan & Li, 2023). For all our experiments, we use a transformer architecture with rotary positions encoding (RoPE) for ILMs and ARMs (Su et al., 2023). For MDMs, we use the DDiT architecture identical to the one used in (Sahoo et al., 2024; Lou et al., 2024). The DDiT is based on the DiT architecture that inserts adaptive layer-norm (AdaLN) in the RoPE based transformer to condition on the time variable (Peebles & Xie, 2023). Since AdaLN has trainable parameters, MDMs with the same hyperparameters as ILMs have slightly more trainable parameters.

## 5.1 PLANNING TASKS

We consider two different planning tasks: the task of generating paths between nodes on a star graph and the task of solving zebra puzzles.

### 5.1.1 STAR GRAPHS

To highlight the key characteristics of the three models, we consider the task of generating the path from a starting node to a target node on star shaped graphs (Bachmann & Nagarajan, 2024). As shown in Figure 3, a star graph is a directed graph with one junction node.

We create three versions of the task. $\text{Star}_{easy}$ only contains symmetric graphs wherein the start node is always the junction node, all paths go out from the junction, and are of equal length. $\text{Star}_{medium}$ and $\text{Star}_{hard}$ contain asymmetric graphs with variable arm lengths, that is, graphs where the start node is not the junction node, there are incoming as well as outgoing edges from the junction node, and most importantly, the arm lengths can be different for each arm of the graph. The easy, medium and hard datasets have graphs with degree 3, 2, 5, respectively, and maximum path length of 5, 6, 12, respectively. We provide an overview of all parameters of the star graphs datasets in Table 4 (Appendix B.0.1). Each graph is presented to the model as a string of edges (expressed as node-pairs) in a random order as shown at the top of Figure 3, where the model needs to predict the path from the start node (green) to the target node (blue). All three models are trained for 50k steps with a learning rate of 1e-4 and batch size of 64. We provide an overview of all hyperparameters in Appendix B.0.1.

For $\text{Star}_{small}$, the optimal autoregressive order of generating the solution is in reverse (target to start) because that makes the dependencies trivial and deterministic. As expected, an ARM trained to predict the path in reverse order gets 100% accuracy on $\text{Star}_{easy}$ as shown in the first row of Table 1. However, it struggles to generate the path in the original left-to-right order (second row) as it requires an implicit lookahead. Since both the MDM and the ILM can generate out-of-order, they get 100% accuracy on $\text{Star}_{easy}$. But the MDM struggles when the lengths of the arms start varying with its sequence level accuracy (seq.) dropping to 36 and 21 on $\text{Star}_{medium}$ and $\text{Star}_{hard}$, respectively. This drop in the performance can be attributed to a deeper limitation of MDMs, which work with absolute token positions. When the arm lengths do not vary, the positions of the junction node and the target node are fixed. However, predicting these positions when the arm lengths vary is intuitively equivalent to solving the puzzle itself in a single pass. ILM continues to perform well in the variable arm length setting because it utilizes relative positions to solve the task iteratively. Some example generation trajectories for ILM are shown in Figure 7 (Appendix C.0.3), where it can be seen that the model tends to start the generation from both ends, leaving the most challenging edges, that is, the junction to latter steps. These results highlight the key advantage of ILMs over MDMs and ARMs: the ability to generate out-of-order while utilizing relative position information. We also implement a single transformer version of the Insertion Transformer (Stern et al., 2019) and compare its performance with the ILM. We find that Insertion Transformer (IT), which uses the EOS token instead of a dedicated stopping classifier like in ILM, consistently undershoots or overshoots the target sequence and therefore performs poorly.[3]

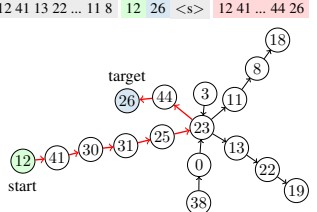

Figure 3: Given the edges of a directed star graph (expressed as a sequence of connected node pairs in a random order), and the start and the target node, the goal is to predict the path from the start to the target node.

Table 1: Exact match accuracy on the star graph and zebra puzzle tasks.

| Model | $\text{Star}_{easy}$ | $\text{Star}_{medium}$ | $\text{Star}_{hard}$ | Zebra |
|-------|------|------|------|------|
| ARMO | 100.0 | - | - | 91.2 |
| ARM | 32.3 | 75.0 | 23.0 | 81.2 |
| MDM | 100.0 | 36.5 | 21.0 | 82.6 |
| IT | 35.2 | 22.1 | 17.5 | - |
| ILM | 100.0 | **100.0** | **99.1** | **90.0** |

---

[3]Qualitative examples for Insertion Transformer are presented in Appendix C.0.2.

| Clue #: | 1 | 2 | 3 | 4 | 5 | 6 |
|---|---|---|---|---|---|---|
| Input: | l((2,2),(2,1)) | =((2,1),(1,2)) | e((1,2)) | =((1,1),(2,2)) | N((2,1),(0,2)) | b((0,1),(1,1),(2,1)) |
| Output: |  | (0,1) (1,0) (2,0) | (0,0) (1,1) (2,2) | (0,2) (1,2) (2,1) | | |
| House #: | | 1 | 2 | 3 | | |

Figure 4: The box contains a compact string representation of a zebra puzzle and its solution. The input is a sequence of constraints in arbitrary order. The solution is a sequence of house,entity, attribute triples, sorted by house number. The complete input output string for this example is given in the Appendix B.0.2.

## 5.2 ZEBRA PUZZLES

Zebra Puzzles are well-known logic puzzles that have been used to benchmark the performance of constraint satisfaction systems (Zebra Puzzle, 2025). The are many variants of Zebra Puzzles, with different sizes and complexity. We use the version introduced in Shah et al. (2024), wherein each puzzle is characterized by a tuple $(m, n)$ where $m$ represents the number of *entities* and $n$ denotes the number of *attributes* associated with each entity. Given some constraints (clues) on the placement of the entity-attribute pairs, the goal is the place each entity-attribute pair in one of the *houses* such that all the constraints are satisfied. Each constraint consists of a *relationship*, and an entity-attribute pair, tuple or triple, for unary, binary, and ternary relationships, respectively. There are 7 types of relationships: = (same house), != (different house), l (left of), L (immediate left), N (neighbor), e, (ends) and b (between). Figure 4 shows an example of a (3,3)-zebra puzzle with 3 entities, 3 attributes, 3 houses, and 6 clues involving the relationships = and l, N, e and b. For the ease of comparison, we use the same setup as well as the same dataset as Shah et al. (2024). We train a 42M parameter transformer model with 8 layers and 8 attention heads with hidden size of 576 with rotary position encoding. The order of solving the constraints plays an important role in the overall performance of the model (Shah et al., 2024). Therefore, to demonstrate the usefulness of out-of-order generation, we train the model on output strings that present the solution in an arbitrary but fixed order that is sorted by house and entity as shown in Figure 4. As seen in the last column of Table 1, the ILM model obtains sequence accuracy of 90% outperforming both the MDM and the ARM, and it even gets close to the performance achieved by the ARM trained on oracle solver decomposed sequence order (Shah et al., 2024).

## 5.3 LANGUAGE MODELING

In order to test the ability of the model to generate short and long text sequences, we pick two small-sized pre-training datasets with different characteristics: (1) The One Billion Word Benchmark (Chelba et al., 2013) (LM1B), and (2) a mixture of TinyStories (Eldan & Li, 2023) and ROC-Stories (Mostafazadeh et al., 2016) (Stories). The LM1B dataset, which has been used to benchmark the performance of MDMs (Austin et al., 2021; Sahoo et al., 2024), consists of short sequences (up to 2-3 sentences) of text from the news domain with a large vocabulary. The TinyStories dataset, on the other hand, consists of 2.1 million stories that 3-4 year old children can understand. In order to increase the diversity of the stories, we also include the ROCStories dataset, which contains 5-sentences stories based on common sense and world knowledge. The combined dataset contains 2.2 million stories in the training set. For both the datasets, we train ILMs, MDMs and ARMs of the same size and architecture (RoPE-based transformer as described above), with $\sim$85M non-embedding trainable parameters (the MDM has slightly more due to the addition of AdaLN layers).[4] We use bert-base-uncased tokenizer for both the datasets and pad each example to 128 tokens for LM1B and 1024 tokens for TinyStories. All the models are trained with an effective batch size of 512, up to 1M steps on LM1B and 60K steps on TinyStories using AdamW (Loshchilov & Hutter, 2019) with a constant learning rate of $10^{-4}$. All the models were trained on 4 A100 (40GB and 80GB) GPUs.

Table 2: Evaluation of unconditional generation quality using per-token NLL under Llama 3.2 3B. The rows with the dataset names contain the NLL and entropy of the examples in the training data.

|  | NLL▼ | Ent▲ | $\overline{\text{len}}$ |
|---|---|---|---|
| Stories | 1.65 | 4.19 | 205 |
| ARM | **2.11** | 4.06 | 201 |
| MDM | 2.54 | 4.55 | 985 |
| ILM (Ours) | 2.14 | 3.76 | 119 |
| LM1B | 3.71 | 3.08 | 28 |
| ARM | **3.94** | 3.12 | 30 |
| MDM | 4.81 | 3.70 | 85 |
| ILM (Ours) | 4.67 | 2.80 | 21 |

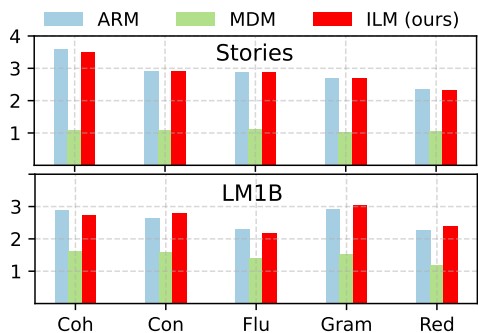

Figure 5: Evaluation of unconditional generation quality using Prometheus 2 7B model as the LLM Judge. Legend: Coh.=coherence, Con.=consistency, Flu.=fluency, Gram.=grammaticality, Red.=non-redundancy.

### 5.3.1 UNCONDITIONAL GENERATION

For sampling unconditional sequences, we use the tau-leaping sampler for the MDM (Sahoo et al., 2024; Campbell et al., 2022) as described in Section 2, and nucleus sampling with $p = 0.9$ for ARM. For ILM, we sample according to Algorithm 2 using two-step ancestral sampling where we first sample the position of insertion using top-k sampling $k \sim p_\theta^{\text{ilm}}(k \mid \boldsymbol{x}[\boldsymbol{b}])$ followed by $v \sim p_\theta^{\text{ilm}}(v \mid \boldsymbol{x}[\boldsymbol{b}], k')$ using nucleus sampling. Our primary metric for evaluating unconditional generation is the per-token negative log-likelihood (NLL) under a large language model and the entropy of the generated text, defined as

$$\text{NLL}(\boldsymbol{x}) = -\frac{1}{|\boldsymbol{x}|} \sum_{i=1}^{|\boldsymbol{x}|} \log p^{\text{LLM}}(x_i | \boldsymbol{x}_{1:i-1}) \quad \text{and} \quad \text{Entropy}(\boldsymbol{x}) = -\sum_{j=1}^{|\mathbb{V}|} c_j \log c_j, \quad (5)$$

where $p^{\text{LLM}}(x_i | \boldsymbol{x}_{1:i-1})$ is the probability of the $i$-th token in the sequence $\boldsymbol{x}$ given the previous $i - 1$, and $c_j = \sum_{i=1}^{|\boldsymbol{x}|} \delta(x_i, v_i)/|\boldsymbol{x}|$ is the relative frequency of the $i$-th vocabulary item $v_i$ in the sequence $\boldsymbol{x}$. We use Llama-3.2-3B (Grattafiori et al., 2024) for computing the NLL. Since NLL and entropy may not be sufficient to judge the overall quality of the generated text, we also use Prometheus 2 7B (Kim et al., 2024) as the LLM Judge to evaluate the quality of the generated text on various linguistic and readability aspects, of which the most important ones are coherence and grammatically (see Appendix B.0.5 for the details of the evaluation prompt).

As seen in Table 2, both the MDM and the ILM obtain worse NLL compared to the ARM trained for the same number of steps, which could be attributed to the training token efficiency and scaling laws for different model types (Nie et al., 2024). However, the ILM performs better than the MDM on both datasets in terms of NLL. In terms of token diversity measured using entropy, the ILM is on the lower side compared to the MDM and the ARM, but still fairly close to the dataset entropy given in the rows with the dataset names. In general, we found that the MDM produces longer sequences than both the ARM, and the ILM, as well as the mean sequence lengths in the training data. We found that to be the main reason for the high entropy (even higher than dataset entropy) of sequences produced by the MDM. The ILM provides linguistically balanced generation similar to ARM and consistently outperforms the MDM, which struggles particularly with coherence and consistency. Notably, MDM's performance deteriorates in the Stories dataset as generation length increases, resulting in more disjointed narratives (see Appendix B.0.6 for the examples). One more difference between the ILM and the MDM is the number of input tokens in each forward pass during inference—for the MDM it stays fixed at maximum allowed sequence length from the beginning, while for the ILM it starts from zero and goes up to the maximum sequence length. Figure 6 shows the impact of per-token generation time on the generation quality measured using per-token NLL under Llama 3.2 3B. For the MDM, we collect samples with varying number of sampling steps (128, 256, 512, and 1024). The generation quality for the MDM (red) improves as per-token generation time/the number of sampling steps is increased, but stays below that of the ILM (blue).

---

[4]Our MDM implementation is based on Sahoo et al. (2024) and it uses log-linear noise schedule.

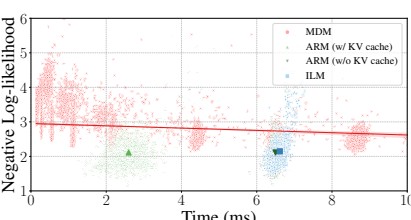

Figure 6: Per-token generation time vs. NLL for the MDM and the ILM trained on the stories dataset.

Table 3: $\Delta$NLL and $\Delta$Entropy denote the percentage change in per-token negative log-likelihood and entropy after infilling, respectively, where subscript gt and inp denote the change with respect to the ground truth and input (sample with the segments removed), respectively.

|  | $\Delta$NLL$_{\text{gt}}$▾ | $\Delta$Ent$_{\text{gt}}$▴ | $\Delta$NLL$_{\text{inp}}$▾ | $\Delta$Ent$_{\text{inp}}$▴ |
|---|---|---|---|---|
| **TinyStories single-segment** | | | | |
| MDM | +14.36 | -3.82 | +3.63 | +1.48 |
| ILM (Ours) | **+12.27** | -4.18 | **+1.79** | +0.04 |
| **LM1B single-segment** | | | | |
| MDM | +25.31 | -0.05 | -0.49 | +4.56 |
| ILM (Ours) | **+20.47** | -1.71 | **-3.57** | +2.64 |
| **LM1B multi-segment** | | | | |
| MDM | +25.64 | +0.15 | -6.02 | +3.97 |
| ILM (Ours) | **+23.52** | -0.79 | **-7.93** | +2.98 |

### 5.3.2 INFILLING

We construct an infilling evaluation dataset by taking 3500 test sequences from the LM1B dataset. The LM1B single-segment dataset is obtained by removing one contiguous segment of tokens from each example, and the multi-segment version is obtained by removing two or more contiguous segments of tokens from each example. Similarly, we construct TinyStories single-segment infilling evaluation dataset by removing the middle sentence from each example from the first 3.3k examples of the TinyStories test dataset.

Since we are evaluating the ability of the pre-trained models to perform arbitrary infilling, we only compare MDMs and ILMs as ARMs are not capable of performing infilling without specialized training. We again employ NLL under Llama-3.2-3B and entropy as the evaluation metrics. However, since we are evaluating the quality of the infilled text, instead of using raw metrics, we use the percentage change $\Delta M_{\text{ref}} = 100 * (M(\boldsymbol{x}) - M(\boldsymbol{x}^{\text{ref}}))/M(\boldsymbol{x}^{\text{ref}})$, where M is either NLL or Entropy, and $\boldsymbol{x}^{\text{ref}}$ is either the input with missing segments (inp) or the ground truth text (gt). Note that when the input text ($x^{\text{inp}}$) is provided to the evaluator LLM, the tokens that belong to the removed segment are completely removed. Therefore, we expect to observe a drop in NLL with respect to the input text and an increase with respect to the ground truth text. As shown in Table 3, we see trends similar to the unconditional generation results. Specifically, the ILM outperforms the MDM on all three evaluation datasets in terms of NLL. On the TinyStories evaluation set, both the MDM and the ILM show an increase in NLL with respect to the input text. However, upon manual inspection, we find that the stories in the dataset are often fairly simple, and removing a sentence from the middle may not change the overall all meaning too much, and hence the NLL for the corresponding input sequences with missing segments is already fairly low.

## 6 DISCUSSION

We explore language modeling by learning to insert tokens and introduce Insertion Language Models (ILMs). We enable successful training of ILMs by using a simple transformer-based parameterization and a denoising objective that approximates a distribution over denoising steps. Using carefully designed synthetic experiments, we demonstrate the failure modes of ARMs and MDMs and show that ILMs overcome these by using out-of-order generation and relative position information. We also demonstrate the usefulness of ILMs for open-ended text generation and arbitrary-length text infilling on medium-sized text corpora.

**Limitations and Future Work.** While ILMs show promising results, in their current form, they still have some limitations. On text data, ILMs still perform slightly worse than ARMs trained for the same number of gradient steps. Using data dependent noising schedule can help close this gap. Similar to MDMs, and unlike ARMs, ILMs also do not allow caching of hidden states and can therefore be slower at inference compared to ARMs with hidden state caching. Addressing these two aspects and scaling ILMs to larger datasets are important directions for future work.

## REPRODUCIBILITY STATEMENT

We provide details about the network datasets, architecture, and training hyperparameters in the empirical evaluation section (Section 5) and the appendix (Section 6).

Anonymized code is available at https://anonymous.4open.science/r/ILMs/README.md.

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

# APPENDIX

## A   EXTENDED RELATED WORK

The exploration of non-autoregressive sequence generation can be traced back to early neural machine translation literature (Ghazvininejad et al., 2019; Stern et al., 2019; Welleck et al., 2019; Gu et al., 2019). But the scaling story of the left-to-right AR LLMs inadvertently diminished the interest in the topic in subsequent years. The success of diffusion models (Sohl-Dickstein et al., 2015; Ho et al., 2020; Song & Ermon, 2019), however, has lead to a resurgence of interest in the topic, but now focusing on scaling in the context of language modeling as opposed to specific sequence-to-sequence tasks like machine translation. There is a vast amount of work on non-autoregressive sequence generation. Here we will try to cover the most relevant works.

**MDMs**   Masked Diffusion Models (MDMs) have been shown to scale competitively to ARMs while addressing some of its key shortcomings (Austin et al., 2021; Campbell et al., 2022; Lou et al., 2024; Sahoo et al., 2024; Shi et al., 2024) on tasks that require planning and following constraints. However, as discussed in Section 2, due to the use of fixed length mask tokens, and simultaneous unmasking, these models, without additional inference time tricks, can generate incoherent sequences. However, as discussed in Section 2, due to the use of fixed length mask tokens, and simultaneous unmasking, these models, without additional inference time tricks, can generate incoherent sequences. To address this, Gong et al. (2024) propose to use a greedy strategy to select the tokens to unmask, Zheng et al. (2024) generalizes it to top-k sampling strategy, while Campbell et al. (2024) utilizes a flow-based formulation to introduce helpful stochasticity on top of the greedy sampling process. All these approaches, rely on inference time techniques to elicit better samples. Ye et al. (2025) modify the MDM training objective by introducing an adaptive token-wise weight that helps the model identify the critical parts of the sequence. This objective, however, is only shown to work for synthetic tasks. Departing from this line of work, we propose a new parameterization and training objective. The MDMs are closely related to order-agnostic sequence models (Yang et al., 2020; Hoogeboom et al., 2021). The key difference between MDMs and order-agnostic models is that unlike MDMs,

which can denoise the entire sequence in one go, order-agnostic models only generate one token at a time in an arbitrary order. Our model also generates the sequence by inserting tokens at arbitrary positions but is allowed to pick the position to insert the token much like Trans-dimensional Jump Diffusion (Campbell et al., 2023), however, unlike Campbell et al. (2023), which is designed for continuous spaces (like videos), we work with discrete space of token sequences. Moreover, we take advantage of the simpler space to instantiate lower variance training objective, which allows us to scale the training to language modeling.

**Other insertion-style models**    There have been several works in the machine translation and early language modeling literature that explore insertion-style models (Gu et al., 2019; Ruis et al., 2020). The Non-monotonic Sequential Text Generation (NMTG) (Welleck et al., 2019) parameterizes an insertion policy. It uses a "learning to search" approach to generate text by inserting tokens to the left or right of the current tokens. While this approach is similar to the ILM, it is comparatively much slower to train due to the high variance of the RL objective. Moreover, the inference process is constrained to be a level-order traversal of a binary tree as opposed to an arbitrary order of insertion, as in the ILM. Due to these two reasons, NMTG is not easily scalable to larger language modeling corpora. The Insertion Transformer (Stern et al., 2019), by virtue of the insertion-based decoding procedure, shares several high-level similarities with the ILM. There are also a few differences, like in the token loss normalization and the decoder architecture. The most significant difference, however, is in the stopping criteria: unlike ILM, the IT does not have a specialized stopping classifier. It instead predicts a special EOS from all slots to decide whether to stop the generation or not. We demonstrate that this approach is unreliable and often overshoots or undershoots the target sequence (see Appendix C.0.2 for a detailed discussion). Stern et al. (2019) also explores the possibility of inserting multiple tokens simultaneously using a fixed binary tree-based insertion scheme. However, we find that insertion of multiple tokens without errors requires context-dependent policy, and leave a detailed exploration of this aspect to future work.

**Infilling**    The ability to insert tokens allow ILMs to perform infilling more naturally compared to ARMs. There has been only a handful of works that focus on the task of arbitrary length infilling using ARMs, most of which require specialized fine-tuning. Bavarian et al. (2022) introduces fill-in-the-middle training objective where ARMs are trained to take `<prefix><suffix>` as the left-context and is required to generate the `<middle>` part such that `<prefix><middle><suffix>` is a meaningful natural language sequence. While this approach enjoys the benefit of adapting an existing pre-trained ARM, its applicability is quite limited because the model is not capable of performing arbitrary infilling, for example, filling two blanks at separate places in the sequence. Gong et al. (2024) also proposes a method to adapt pre-trained ARMs to masked denoising models. However, once adapted, the model has the same limitations as MDMs.

**Shortcomings of left-to-right generation.**    There are several works that attempt to study the shortcomings of left-to-right sequence generation using controlled experiments on synthetic tasks (Bachmann & Nagarajan, 2024; Frydenlund, 2024; 2025). Bachmann & Nagarajan (2024) show that left-to-right generation using next-token prediction training paradigm has problems when there are some tokens that are much harder to predict than others. Frydenlund (2024) show that the star-graph task with fixed arm lengths can be solved using teacher-forcing but with modified input ordering where the edges in the input are not shuffled, making the task somewhat trivial. Frydenlund (2025) show that the pathological behaviour for next-token prediction paradigm on star-graph task is due to excessive supervision for "easy" prediction steps, i.e., the steps that follow the "hard" step of junction node. MDMs circumvent this issue of excessive supervision by trying to predict all the tokens simultaneously. This introduces, so called, task decomposition (Frydenlund, 2025; Kim et al., 2025b). In our work, we generalize the star-graph task to incorporate variable arm lengths, and show that while MDMs can induce task decomposition when the output sequence lenghts are fixed, but struggle with variable sequence lengths.

**Concurrent work**    Havasi et al. (2025) proposes to train a transformer to perform insertion, deletion and substitution and train it using flow matching (Campbell et al., 2024) objective. Similarly, Kim et al. (2025a) utilizes the stochastic interpolant framework (Albergo et al., 2023) to formulate

insertion-based MDM, wherein mask tokens are progressively inserted and then filled in subsequent generation steps.

## B  EXPERIMENTAL DETAILS

### B.0.1  STAR GRAPHS

All three models, the ILM, the MDM and the ARM, are RoPE-based transformers with $\tilde{8}4$M parameters with 12 attention heads and 12 layers with hidden size of 768.

Table 4: Different star graph datasets used in the experiments. All the datasets use asymmetric graphs, meaning the start and the goal nodes both are away from the junction, and the target path passed through the junction. VStar version additionally has variable arm lengths $a$ in the same input star graph.

| Name | Degree | $\min(a)$ | $\min(l)$ | $\max(l)$ | $|\mathbb{V}|$ | #Train | #Test |
|---|---|---|---|---|---|---|---|
| $\text{Star}_{\text{easy}}$ | 3 | 1 | 5 | 5 | 20 | 50k | 5k |
| $\text{Star}_{\text{medium}}$ | 2 | 2 | 3 | 6 | 20 | 50k | 5k |
| $\text{Star}_{\text{hard}}$ | 5 | 5 | 6 | 12 | 56 | 50k | 5k |

### B.0.2  ZEBRA PUZZLES

We use the dataset created by Shah et al. (2024), which they make publicly available at zebra train and zebra test. The train dataset contains about 1.5 million puzzles and the test set contains about 100 thousand puzzles. Following the experimental setup in Shah et al. (2024), we train for 500k steps after which the change in training loss is negligible. Table 5 shows an example input and output from the dataset.

| (m,n) | Inputs | Outputs |
|---|---|---|
| (3,3) | left-of LHS c 2 2 RHS c 2 1 CLUE_END = LHS c 2 1 RHS c 1 2 CLUE_END ends LHS c 1 2 RHS CLUE_END = LHS c 1 1 RHS c 2 2 CLUE_END nbr LHS c 2 1 RHS c 0 2 CLUE_END inbetween LHS c 0 1 RHS c 1 1 c 2 1 CLUE_END | 0 0 1 1 0 0 2 0 0 0 1 2 1 1 1 2 1 2 0 2 0 1 2 2 2 2 1 |
| | Vocab: 0, 1, 2, 3, 4, 5, nbr, left-of, inbetween, immedate-left, end, !=, =, CLUE_END, RHS, LHS | |

Table 5: Example inputs and outputs for the zebra puzzles. Each example is a concatenation of the input and output strings. The strings are tokenized using space and the tokenizer uses a custom vocabulary as shown in the table. The output string is entity-house-attribute.

### B.0.3  LANGUAGE MODELING: STORY GENERATION

We combine the TinyStories (Eldan & Li, 2023) and ROCStories (Mostafazadeh et al., 2016) datasets. The combined dataset contains almost 2.2 million stories (2,198,247) in the training set. We use randomly selected 3.3k stories from the test split for performing infilling evaluation. The stories were generate using GPT-3.5 and GPT-4. TinyStories has longer sequences but a smaller vocabulary compared to LM1B.

### B.0.4  LANGUAGE MODELING: LM1B

We use a model with 85M parameters, consisting of 12 layers and 12 attention heads, trained with a learning rate of 0.0001 for 1M steps.

### B.0.5 LLM EVALUATION USING PROMETHEUS-2

We use Prometheus-2 7B model and follow the evaluation protocol given in Kim et al. (2024). For evaluating natural language generation, we use metrics like: Coherence, Consistency, Fluency, Grammaticality, Non-Redundancy and Spelling Accuracy. We generate evaluation text using a sampling temperature of 0.0, a maximum token limit of 1k, and a top-p value of 0.9

**LLM-As-Judge Evaluation Prompt:**

> You are a fair judge assistant tasked with providing clear, objective feedback based on specific criteria, ensuring each assessment reflects the absolute standards set for performance.
>
> **Task Description:**
> An unconditional generation to evaluate, and a score rubric representing an evaluation criteria are given.
> 1. Write a detailed feedback that assesses the quality of the generation strictly based on the given score rubric, not evaluating in general.
> 2. After writing a feedback, write a score that is an integer between 1 and 5. You should refer to the score rubric.
> 3. The output format should look as follows: `"(write a feedback for criteria) [RESULT] (an integer number between 1 and 5)"`.
> 4. Please do not generate any other opening, closing, or explanations.
>
> **Generation to evaluate:**
> `{generation}`
>
> **Score Rubrics:**
> `{rubrics}`
>
> **Feedback:**

| Rubric Item | Rubric Text |
|---|---|
| Coherence | (Is the text coherent and logically organized?)
Score of 1: Very incoherent. The generation lacks structure, has sudden jumps, and is difficult to follow.
Score of 2: Somewhat incoherent. The generation has some semblance of structure, but has significant flaws in flow and organization.
Score of 3: Neutral. The generation is decently organized, with minor issues in flow and structure.
Score of 4: Mostly coherent. The generation is well-structured with very few minor coherence issues.
Score of 5: Highly coherent. The generation is excellently organized, flows seamlessly, and builds information logically from start to end. |
| Consistency | (Is the text consistent in terms of style, tone, and tense?)
Score of 1: The text is inconsistent in style, tone, and tense, leading to confusion.
Score of 2: The text shows occasional inconsistencies in style, tone, and tense.
Score of 3: The text is mostly consistent in style, tone, and tense, with minor lapses.
Score of 4: The text is consistent in style, tone, and tense, with rare inconsistencies.
Score of 5: The text is highly consistent in style, tone, and tense throughout. |
| Fluency | (Is the text fluent and easy to read?)
Score of 1: The text is disjointed and lacks fluency, making it hard to follow.
Score of 2: The text has limited fluency with frequent awkward phrasing.
Score of 3: The text is moderately fluent, with some awkward phrasing but generally easy to follow.
Score of 4: The text is fluent with smooth transitions and rare awkward phrases.
Score of 5: The text is highly fluent, with natural and smooth expression throughout. |
| Spelling Accuracy | (Does the text demonstrate correct spelling?)
Score of 1: The text contains frequent spelling errors, making it difficult to understand.
Score of 2: The text has multiple spelling errors that affect readability and clarity.
Score of 3: The text has occasional spelling errors, but they do not significantly impact comprehension.
Score of 4: The text is mostly free of spelling errors, with only rare mistakes that do not affect understanding.
Score of 5: The text has perfect spelling accuracy, with no errors present. |
| Grammaticality | (Does the text demonstrate proper grammatical usage?)
Score of 1: The text contains frequent grammatical errors, making it difficult to understand.
Score of 2: The text shows occasional grammatical errors, which disrupt the flow and clarity of the text.
Score of 3: The text generally adheres to grammatical rules, though minor errors are present.
Score of 4: The text demonstrates good grammaticality with rare errors that do not affect comprehension.
Score of 5: The text excels in grammatical usage, with clear and correct grammar throughout. |
| Non-Redundancy | (Does the text avoid unnecessary repetition?)
Score of 1: The text is highly redundant, with excessive repetition of words, phrases, or ideas that make it difficult to read.
Score of 2: The text contains noticeable redundancy, with multiple instances of unnecessary repetition that affect clarity.
Score of 3: The text has some minor redundancy, but it does not significantly impact readability or meaning.
Score of 4: The text is mostly nonredundant, with rare instances of repetition that do not affect clarity.
Score of 5: The text is highly concise and avoids any unnecessary repetition, presenting ideas efficiently and effectively. |

Table 6: Rubric for NLG Evaluation through LLM-As-Judge

### B.0.6 UNCONDITIONAL GENERATION EXAMPLES

Below we provide unconditional generations from all the models, along with the evaluation scores generated by the LLM judge as explained in appendix B.0.5.

**Dataset: Stories — Model: ARM — Overall Score: 1**

lily loved ice – cream. she liked to eat ice – cream when it was cold and sweet. one day, she saw a big bowl of ice – cream on the table. she wanted to eat it all. she took a big spoon and scooped some ice – cream into her mouth. but the ice – cream was not good. it was icy and sticky and tasted bad. lily spat it out and made a face. she felt sick and her mouth hurt. she started to cry. she ran to her mom and said, " mom, mom, i ate ice – cream. it was bad and cold and tasted good. the ice – cream fell on the floor and it made me sick. " her mom hugged her and said, " i ' m sorry, lily. you didn ' t do the chore. you had ice – cream for dinner. it was not good. it made your mouth hurt and your tummy ache. you have to listen to me and do the answer. " lily nodded and said, " i ' m sorry, mom. i wanted to eat ice – cream. but it was too bad. it made me sick and i ate something bad. can i have some water, please? " her mom smiled and said, " of course, sweetie. here you go. feel better. and guess what? i have a surprise for you. look! " she took a plate from the cabinet and said, " i scooped some ice – cream every day for you. it was still cold and fresh and useful. see? " she pointed to the plate. lily saw the ice – cream. she was happy and relieved. she said, " wow, mom, you made the ice – cream for me? it looks delicious. thank you, mom. can i have some ice – cream now? " her mom said, " yes, you can. but you have to throw it away first. and you have to give it to me. the timer is off. " lily did as her mom said. she threw away the ice – cream and said, " ok, mom. i will do it. i like ice – cream. but i won ' t eat ice – cream again. and i won ' t use the cold. it ' s bad and i want to make you happy. " her mom said, " i ' m proud of you, lily. you are a smart and sweet girl. you made me happy. but you also made me sad. the ice – cream does not make you well. it gives me energy and i want to enjoy it. it also gives me love and hugs and kisses. it ' s good for me and for lily. it makes me happy too. do you want some water and milk now? " lily said, " yes, please. i want some water and milk. and some ice – cream. thank you, mom. i love you and i love the ice – cream. but i don ' t like it. i don ' t like getting sick sometimes. it makes me sad too. " they went to the kitchen and drank some water and milk. they talked and laughed and watched the sun go down and make the air warm and clear. they cuddled on the couch and watched the sunset. they were happy and safe. they were no longer sad. they were good.

**Dataset: Stories — Model: ARM — Overall Score: 5**

it was a magnificent night. jill decided to take a walk around the neighborhood. she saw a group of children playing in the park. they were having so much fun. they were all gossipling and laughing.

**Dataset: Stories — Model: MDM — Overall Score: 1**

ben liked to help his mom with animals. he had cows and chickens and sheep, and sheep, and hay. he liked to feed him wooly with his glass and play with it. " hello, sheep, ben. you are the best helper in the farm, " wooly cooed and wagged his tail. anna showed her his bowl of bread and gave her a small bowl. " i think so, ben, you can have some of his favorite. you can feed him his milk with him, " his mom said, sharing the bread with him. ben smiled and ate the bread carefully. the dog licked his face and wagged his tail. it was soft and friendly spot. it was not the petter, but he belongs to a cowy, but she lived nearby. " can we go to the farm with her? " ben asked, curious. " no, ben, spot belongs to the wild spot in his barn. he knows not to come back soon. he is just playing with us. she is not shy, but she is very nice. come on on, let ' s go play with her in the barn, " she said. ben nodded and went to the barn spot with his mom. he liked all the animals and plants. he opened the window and called his mom, " ben, you have to be quiet and gentle. you can break a hole easily. and you can pet the cow or moo, " she said. ben looked at oinky and tilted his head. he was afraid of oinky. he wanted some beef or carrots. he thought mom was lonely. " mom, i want to find out, " he said. " maybe they are not scary. maybe there are animals in the farm. " ben peeked inside. he hoped there were a toy, or a car, or a toy car. he saw ducks, frogs, and the farm. he looked around and saw a big furry animal with a hat and a coat. he thought, " maybe it is the cow or moo. " moo looked at him with his eyes. he seemed friendly, like, " hello, cowy what are you doing here? " " doo, mooing, " ruo replied. sara looked surprised. she was surprised. she knew ben had gone to the sack of food. ben hadn ' t seen the cow or the pig. he had never been able to eat them. they were very nice and friendly. please, mom, please, come and see, " he asked, begging sara to come out again. he reached for his mom to oinky, but his mom wasn ' t mad. she said, " no, ben, stop. he might be hungry. and it is too cold for you. come on, and let ' s go home for lunch. you should not go to anything about him. " sara want to oinky afraid. he seemed nice and soft. she put a box next to her bed. she whispered, " maybe i can ' t touch him again. " ben did not listen. he reached the cow and got up. he did not see a cut on his shirt and his tooth. and he behind him and s cold and hard. ouch! ben fell down. he landed on the floor and bumped into something. it hurt a lot. sara ' s mom heard ben ' s cry and ran to check on him. she saw ben on the floor looking sad. she ran to him and said, " i ' m sorry, i ' m sorry ben. she ' s not mad at you. can you see her now? her finger hurts? " ben said, " no, i ' m not okay. she ' s just blood on her finger. i held her leg and said, " ow, mom. that ' s my cow. ' " his mom said, " don ' t worry, ben. you saved me. you ' re not brave and strong. but, i ' m lucky i tried to help you. but not. now come on. let ' s go home. you will be okay. " she did not. she knew they were going to the doctor. she took the bandage out of the sack and cut it seped. she gave it to sara and said, " here ben, i ' m here you. i love you. i ' m glad you like cowy, okay. when mom arrived, ben saw sara waiting for help. he told her they were sorry, but mom was still angry or embarrassed. she hugged her and said, " i ' m so happy for you, ben. you should calm down and a good sister. you have a great mom. don ' t you feel to forgive him and me? " ben hugged mom and said, " thank you, mom. i forgive. " they both smiled. their mom was proud too. they were glad. they kissed ben and kissed him. they also said, " sara, and so is tom

**Dataset: Stories — Model: MDM — Overall Score: 3.6**

once upon a time, there was a brave monkey named timmy. timmy loved to climb up in the tree in the jungle. one day, timmy met a scary lion. the lion looked sad and lonely. timmy knew he had to help his friend and make him feel better. timmy decided to follow the lion back home. when the lion arrived at its den, the lion said, " we told you, we can still be friends. " timmy was so happy for being brave and said he you back to the lion said, " you ' re welcome. " timmy and the lion became the best of friends. the lion became a brave friend and they played together in the jungle every day.

**Dataset: Stories — Model: ILM — Overall Score: 1**

once upon a time there was a box. it was a special box. one day it wanted to go somewhere. it asked if it was ok, so it started to move. and soon, the box was ready! it was so fun. the box danced and laughed and smiled. they were so happy that they stayed in the box forever.

**Dataset: Stories — Model: ILM — Overall Score: 4**

```
once upon a time, there was a little girl named lily.  she was very curious about the world around
her.  one day, she decided to pack up her toys and go to the park.  but as she was packing her
things, she saw a big rock.  she knew the rock was not safe, so she decided to leave the rock alone.
when she got home, she told them about the rock.  her family was very upset and told her it was not
safe to play with rocks.  from that day on, lily never played with anything else again.  the end.
```

**Dataset: LM1B — Model: ARM — Overall Score: 1**

```
for me, the life a doctor receives is what he is doing.
```

**Dataset: LM1B —Model: ARM — Overall Score: 5**

```
i think you will find a lot more talent than you may have.
```

**Dataset: LM1B — Model: MDM — Overall Score: 1**

```
hazex ga, pixi ( ebookcinecon.  com ) and 1 ) apply exclusive control over the world to theguardit
and the inu digital tv device which allows viewers to view hd e2, with itv more than ( instead of
dvds using hd ) 3 :  and thaw kept ".  " the information if possible using digital ' s most erasable
delivery configuration software.  atusa vip technology could also facilitate the use of add – print
anywhere while handling unique customer experiences.  the content of exorult manage and / or the
donetv is natural and feature top hits.  pacelle also licenses all content and ommi
```

**Dataset: LM1B — Model: MDM — Overall Score: 3.8**

```
a third of four blackers headteachers report regular use of cannabis with alcohol levels, according
to a study published in scientific paper.
```

**Dataset: LM1B — Model: ILM — Overall Score: 1**

```
at that moment, he could be the next great medical doctor, so he or she died.
```

**Dataset: LM1B — Model: ILM — Overall Score: 5**

```
there were no casualties or injuries in the violence.
```

## C  ADDITIONAL RESULTS AND EXAMPLES

### C.0.1  TOKEN ACCURACY ON STAR GRAPHS AND ZEBRA PUZZLES

Table 7: Performance (in terms of accuracy) on the star graph planning task.

| Model | Star$_{easy}$ | | Star$_{medium}$ | | Star$_{hard}$ | | Zebra |
|---|---|---|---|---|---|---|---|
| | Sequence Acc. | Token Acc. | Sequence Acc. | Token Acc. | Sequence Acc. | Token Acc. | Sequence Acc. |
| ARMO | 100.0 | 100.0 | - | - | - | - | 91.2 |
| ARM | 32.3 | 81.7 | 75.0 | 81.4 | 23.0 | 43.2 | 81.2 |
| MDM | 100.0 | 100.0 | 36.5 | 90.6 | 21.0 | 54.9 | 82.6 |
| IT | 35.2 | 98.2 | 22.1 | 80.9 | 17.5 | 79.9 | - |
| ILM | 100.0 | 100.0 | **100.0** | **100.0** | **99.1** | **99.7** | **90.0** |

### C.0.2  COMPARISON WITH INSERTION TRANSFORMER

The Insertion Transformer (IT) (Stern et al., 2019) differs from ILM in the following ways:

1. The IT is an encoder-decoder model, while the ILM is a decoder-only model.

2. The IT uses a specialized final layer on top of a transformer decoder, while the ILM uses a standard transformer decoder architecture.

3. The IT uses local averaging for token prediction loss (equation 14 in Stern et al. (2019)), i.e., the denominator is the number of tokens in the ground truth for a particular slot, while we use the global average in Equation (2) wherein the numerator is a single sum of the negative log-likelihood corresponding to all missing tokens and the denominator is the total number of missing tokens in all the slots combined.

4. The IT does not have a specialized stopping classifier. It instead predicts a special EOS from all slots to decide whether to stop the generation or not.

Using (2) and (3) in our setting yields an informative ablation. Therefore, we implement a decoder-only Insertion Transformer using the same transformer architecture as the ILM but with the loss provided in Stern et al. (2019). As seen in the Table 1, the IT performs poorly compared to the ILM on the star graphs task. Upon qualitative inspection, we find that IT, which uses the EOS token instead of a dedicated stopping classifier like in ILM, consistently undershoots or overshoots the target sequence. Due to this, its sequence accuracy is substantially lower than token accuracy. Below, we present two examples from the validation set that illustrate the issue.

```
Input:
10 17 15 4 19 6 17 1 4 12 1 16 4 19 9 10 8 5 0 8 6 3 7 4 4 9 12 0 7 5 
Predicted Output:
7 4 12 0 8 5
Target Output:
7 4 4 12 12 0 0 8 8 5

Input:
14 11 9 6 7 9 11 19 19 16 17 3 3 11 16 5 11 1 11 7 1 10 11 10 
Predicted Output:
11 1 1 10 11 11 1 1 10
Target Output:
11 1 1 10
```

### C.0.3   STAR GRAPHS

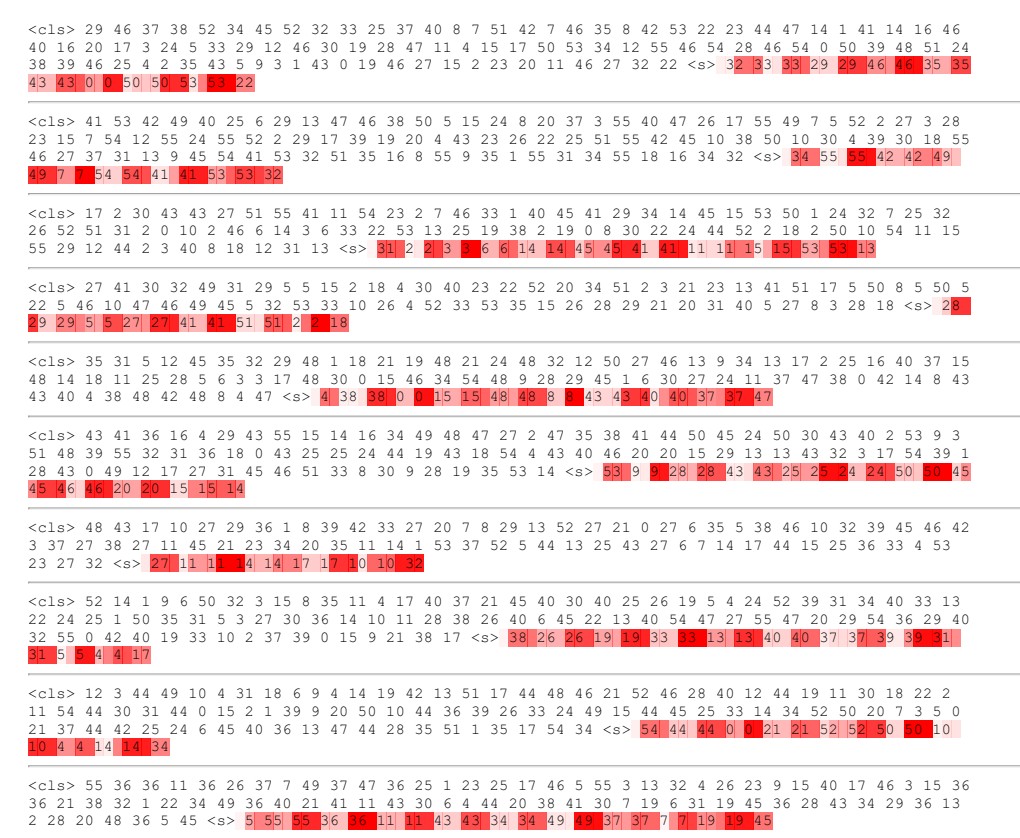

Figure 7: Generation trajectories for ILM on 10 test examples from the Star_hard task. Lighter color indicates that the token was generated earlier than the ones with the darker color.

### C.0.4 LANGUAGE MODELING: UNCONDITIONAL GENERATION TRAJECTORIES

In general, we observe that the ILM's generations are shorter than the MDM's, which is one of the reasons for higher coherence scores for ILM. The MDM, due to its longer generations, messes up entities and long range consistency more frequently. Repetition manifests in ILM's generations differently due to its ability to insert. Specifically, we observe that there is a pattern that generations have some alliterations, like short phrases "he put", "they run", etc. appear at many locations in the same generation. We believe that this is due the token frequencies in the Stories data set. These simple phrases are quite frequent in the dataset and therefore ILM inserts many of them early in the generation process and then fills in more nuanced tokens as shown in the generation trajectory in the last example in this section. We also find that ILM's generations have lesser grammatical mistakes than MDM as shown in our LLM judge evaluation in fig. 5.

**Generation Sequence (LM1B)**

```
Step 1:
Step 1:   the
Step 2:   the.
Step 3:   the saturday.
Step 4:   the in saturday.
Step 5:   the in to saturday.
```

```
Step 6:   the in to begin saturday.
Step 7:   the in is to begin saturday.
Step 8:   the mission in is to begin saturday.
Step 9:   the mission in colombo is to begin saturday.
Step 10:  the mission in colombo is scheduled to begin saturday.
Step 11:  the mission in colombo is scheduled to begin later
saturday.
```

**Generation Sequence (LM1B)**

```
Step 1:
Step 1:  '
Step 2:  '  s
Step 3:  '  s were
Step 4:  that ' s were
Step 5:  that ' s were he
Step 6:  " that ' s were he
Step 7:  " that ' s were the he
Step 8:  " that ' s were the " he
Step 9:  " that ' s were the " he.
Step 10:  " that ' s what were the " he.
Step 11:  " that ' s what were the of " he.
Step 12:  " that ' s what were the of, " he.
Step 13:  " that ' s what were the of, " he says.
Step 14:  " that ' s what we were the of, " he says.
Step 15:  " that ' s what we were the target of, " he says.
Step 16:  " that ' s what we were the target of today, " he says.
Step 17:  " that ' s what we were on the target of today, " he says.
Step 18:  " that ' s what we were on the target of early today, " he
says.
```

**Generation Sequence (LM1B)**

```
Step 1:
Step 1:  .
Step 2:  he.
Step 3:  of he.
Step 4:  the of he.
Step 5:  the of world he.
Step 6:  the the of world he.
Step 7:  the the of world " he.
Step 8:  the economic the of world " he.
Step 9:  the economic the of in world " he.
Step 10:  the economic down the of in world " he.
Step 11:  the economic down the of economy in world " he.
Step 12:  the economic down the of economy in world, " he.
Step 13:  the economic down the and of economy in world, " he.
Step 14:  the economic down in the and of economy in world, " he.
Step 15:  the economic down in the and a of economy in world, " he.
Step 16:  the economic down in the and a of economy in the world, "
he.
```

```
Step 17:  the economic downturn in the and a of economy in the
world, " he.
Step 18:  the economic downturn in the and a of economy in the world
economy, " he.
Step 19:  we the economic downturn in the and a of economy in the
world economy, " he.
Step 20:  we the economic downturn in the and a of the economy in
the world economy, " he.
Step 21:  we the economic downturn in the and have a of the economy
in the world economy, " he.
Step 22:  we the economic downturn in the and we have a of the
economy in the world economy, " he.
Step 23:  we the economic downturn in the world and we have a of the
economy in the world economy, " he.
Step 24:  we are the economic downturn in the world and we have a of
the economy in the world economy, " he.
Step 25:  we are the economic downturn in the world and we have a of
the economy in the world economy, " he said.
Step 26:  we are the economic downturn in the world and we have a
share of the economy in the world economy, " he said.
Step 27:  we are fighting the economic downturn in the world and we
have a share of the economy in the world economy, " he said.
Step 28:  we are fighting the economic downturn in the world and we
have a fair share of the economy in the world economy, " he said.
Step 29:  we are fighting the worst economic downturn in the world
and we have a fair share of the economy in the world economy, " he
said.
Step 30:  " we are fighting the worst economic downturn in the world
and we have a fair share of the economy in the world economy, " he
said.
```

**Generation Sequence (LM1B)**

```
Step 1:
Step 1:  .
Step 2:  '.
Step 3:  '  is.
Step 4:  that ' is.
Step 5:  that ' is on.
Step 6:  that ' the is on.
Step 7:  that ' s the is on.
Step 8:  that ' s the is taking on.
Step 9:  that ' s the administration is taking on.
Step 10:  that ' s the current administration is taking on.
Step 11:  that ' s what the current administration is taking on.
Step 12:  that ' s what the current administration is taking on it.
Step 13:  that ' s what the current administration is taking on it
now.
Step 14:  that ' s what the current bush administration is taking on
it now.
Step 15:  that ' s what the current bush administration is taking on
it right now.
Step 16:  but that ' s what the current bush administration is
taking on it right now.
```

**Step 17:** but that ' s not what the current bush administration is taking on it right now.

**Step 18:** but that ' s not like what the current bush administration is taking on it right now.

**Step 19:** but that ' s not entirely like what the current bush administration is taking on it right now.

**Step 20:** but that ' s not entirely like what the current bush administration is taking on it all right now.

## Generation Sequence (Stories)

**Step 1:**

**Step 1:** .

**Step 6:** a. the it his to

**Step 9:** there a. the the and it his to

**Step 17:** there a he. the he was the. the and the it and his. to

**Step 26:** there was a he to he. he the. he was he the. he the and the a it the and his. to

**Step 27:** there was a he to he. he the. he was he the. he the and the a it the and his. he to

**Step 28:** there was a he he to he. he the. he was he the. he the and the a it the and his. he to

**Step 29:** there was a he he to he. he the. he was he the. he the and the a he it the and his. he to

**Step 49:** upon there was a man he to something, he to. he found. so excited. he put it the. he was. he the. he put the in and the the a he the it the man smiled and he his. he to

**Step 65:** once upon there was a man he wanted to buy something, so he went to buy a. he found a. was so excited. he put it the. he was. he in the. he put the in his and the he the a he the it. the man smiled and he his. he was to have the

**Step 66:** once upon there was a man he wanted to buy something, so he went to buy a. he found a. was so excited. he put it the. he was. he in the. he put the in his and the he the a he the it. the man smiled and he his. he was to have the.

**Step 110:** once upon a time there was a wealthy man. he wanted to buy something, so he went to buy a bed. he found a beautiful bed. he was so excited. he put it in the shop. he was very happy. he went home and put it in the house. he put the bed in his room and opened the door. then he put the bed on with a cozy blanket. he put the bed down and enjoyed it. the man smiled and he put the blanket on his bed. he was so happy to have the bed.

**Step 139:** once upon a time, there was a wealthy man. he wanted to buy something special, so he went out to buy a special bed. he found a beautiful, pink bed. he was so excited. he put it in his pocket, and the shopkeeper smiled. he was very happy. he went home and put it in the house. he put the bed in his living room, and then he opened the bedroom door. then, he put the bed on the bed with a cozy blanket. then, he put the bed down and enjoyed it. the wealthy man smiled and laughed as he put the special pink blanket on his bed. he was so happy to have made the perfect bed.

**Step 143:** once upon a time, there was a wealthy man. he wanted to buy something very special, so he went out to buy a special bed. he found a beautiful, pink bed. he was so excited. he put it in his pocket, and the shopkeeper smiled. he was very happy. he went home and put it in the house. he put the bed in his living room, and then he opened the bedroom door. then, he put the bed on the bed with a cozy, pink blanket. then, he put the bed down and enjoyed

```
it.  the wealthy man smiled and laughed as he put the special, pink
blanket on his bed.  he was so happy to have made the perfect bed.
```

### C.0.5 CONDITIONAL GENERATION: INFILLING

Below we compare the outputs of the ILM and the MDM for variable length infilling task. ILM has the freedom to implicitly decide the length of the infilled segment. We observe that ILM generally chooses to insert shorter segments compared the ground truth.

**Dataset: Stories**

**Input** once upon a time, in a big forest, there lived a rhinoceros named roxy.  roxy loved to climb. ____ one day, roxy found an icy hill.  she had never seen anything like it before.  it was shiny and cold, and she wanted to climb it.  roxy tried to climb the icy hill, but it was very slippery.  she tried again and again, but she kept falling down.  roxy was sad.  ____ billy saw that roxy was sad and asked, " why are you sad, roxy? " roxy told billy about the icy hill and how she couldn ' t climb it.  billy said, " i have an idea!  let ' s find some big leaves to put under your feet.  they will help you climb the icy hill.  " roxy and billy looked for big leaves and found some.  roxy put the leaves under her feet and tried to climb the icy hill again.  this time, roxy didn ' t slip.  she climbed and climbed until she reached the top of the icy hill.  roxy was so happy!  ____ from that day on, roxy and billy were the best of friends, and they climbed and played together all the time.  and roxy learned that with a little help from a friend, she could climb anything.

**MDM**

once upon a time, in a big forest, there lived a rhinoceros named roxy.  roxy loved to climb. she had a big ball in her mouth. one day, roxy found an icy hill.  she had never seen anything like it before.  it was shiny and cold, and she wanted to climb it.  roxy tried to climb the icy hill, but it was very slippery.  she tried again and again, but she kept falling down.  roxy was sad. she started to feel distant.  suddenly, she felt a tap on her ankle.  it was billy. billy saw that roxy was sad and asked, " why are you sad, roxy? " roxy told billy about the icy hill and how she couldn ' t climb it.  billy said, " i have an idea!  let ' s find some big leaves to put under your feet.  they will help you climb the icy hill.  " roxy and billy looked for big leaves and found some. roxy put the leaves under her feet and tried to climb the icy hill again.  this time, roxy didn ' t slip.  she climbed and climbed until she reached the top of the icy hill.  roxy was so happy! she hugged billy, and thanked him for helping her. from that day on, roxy and billy were the best of friends, and they climbed and played together all the time.  and roxy learned that with a little help from a friend, she could climb anything.

**ILM**

once upon a time, in a big forest, there lived a rhinoceros named roxy.  roxy loved to climb.  one day, roxy found an icy hill.  she had never seen anything like it before.  it was shiny and cold, and she wanted to climb it.  roxy tried to climb the icy hill, but it was very slippery.  she tried again and again, but she kept falling down.  roxy was sad.  her friend billy saw that roxy was sad and asked, ẅhy are you sad, roxy? ṙoxy told billy about the icy hill and how she couldn ' t climb it.  billy said, ï have an idea!  let ' s find some big leaves to put under your feet.  they will help you climb the icy hill.  ṙoxy and billy looked for big leaves and found some.  roxy put the leaves under her feet and tried to climb the icy hill again.  this time, roxy didn ' t slip.  she climbed and climbed until she reached the top of the icy hill.  roxy was so happy!  from that day on, roxy and billy were the best of friends, and they climbed and played together all the time.  and roxy learned that with a little help from a friend, she could climb anything.

**Dataset: Stories**

**Input**

one day, a small cat named tim was walking in the park.  the sun was very bright and the wind was harsh.  ____" do you want to play?  šam looked at tim and said, ÿes!  let ' s play!  ẗhey played all day, but then tim hurt his foot.  ____ sam wanted to help his friend feel better.  sam had an idea. he offered tim a ride on his back.  ï can carry you home, šaid sam.  tim smiled and said, ẗhank you, sam!  ẗhey went home together, and tim ' s foot felt better soon.  the next day, they played again, and they were both very happy.

**MDM**

"one day, a small cat named tim was walking in the park.  the sun was very bright and the wind was harsh.  sam was playing faster than him.  ḧi, welcome to my!  šaid the mouse.  ḋo you want to play? šam looked at tim and said, ÿes!  let ' s play!  ẗhey played all day, but then tim hurt his foot. he had fallen in the grass and it hurt a lot. sam wanted to help his friend feel better.  sam had an idea.  he offered tim a ride on his back.  ï can carry you home, šaid sam.  tim smiled and said, ẗhank you, sam!  ẗhey went home together, and tim ' s foot felt better soon.  the next day, they played again, and they were both very happy."

**ILM**

"one day, a small cat named tim was walking in the park.  the sun was very bright and the wind was harsh.  tim asked his friend sam, ḋo you want to play?  šam looked at tim and said, ÿes!  let ' s play!  ẗhey played all day, but then tim hurt his foot.  sam wanted to help his friend feel better. sam had an idea.  he offered tim a ride on his back.  ï can carry you home, šaid sam.  tim smiled and said, ẗhank you, sam!  ẗhey went home together, and tim ' s foot felt better soon.  the next day, they played again, and they were both very happy.,

## Dataset: LM1B

**Input**

i would not be upset to see criminal charges brought against them as well as they were endangering the lives of more than 100 ___ eighteen minutes and could have gotten everyone aboard killed, plus people on the ground.  losing their licenses is too mild a punishment.

**MDM**

i would not be upset to see criminal charges brought against them as well as they were endangering the lives of more than 100 iowa farm workers, the firefighters,.  the fire lasted just eighteen minutes and could have gotten everyone aboard killed, plus people on the ground.  losing their licenses is too mild a punishment.

**ILM**

i would not be upset to see criminal charges brought against them as well as they were endangering the lives of more than 100 passengers in eighteen minutes and could have gotten everyone aboard killed, plus people on the ground.  losing their licenses is too mild a punishment.

## Dataset: LM1B

**Input**

tony rutherford, chairman of the british fertility society, welcomed the birth, but added :  ït ___ research., and should only be offered to patients within the context of a robustly designed clinical trial, carried out in suitably experienced centres.

**MDM**

tony rutherford, chairman of the british fertility society, welcomed the birth, but added :  ït is also a good example of the benefit highlighted in this central research., and should only be offered to patients within the context of a robustly designed clinical trial, carried out in suitably experienced centres.

**ILM**

tony rutherford, chairman of the british fertility society, welcomed the birth, but added :  ït was important for cancer research., and should only be offered to patients within the context of a robustly designed clinical trial, carried out in suitably experienced centres.

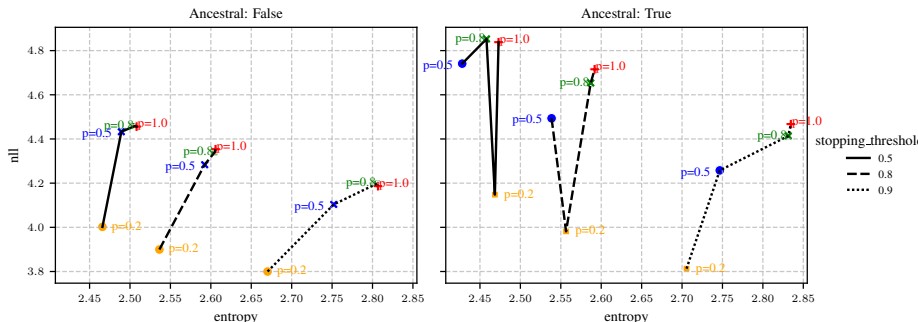

Figure 8: The figure shows NLL vs. entropy on LM1B as the sampling parameters are varied for ILM. A conservative stopping threshold of 0.9 provides a good balance between quality and diversity.

## C.1 ANALYSIS OF SAMPLING HYPERPARAMETERS

Figure 8 shows the impact of various sampling parameters on the quality and diversity of the generations. A conservative stopping threshold of 0.9 provides a good balance between quality (lower nll) and diversity (higher entropy).

## D CONNECTION BETWEEN ILM AND DISCRETE DENOISING

Consider a discrete time markov chain $(X_t)$ with states taking values in $\mathbb{V}^L$ with the transition kernel $q(X_t \mid X_{t-1})$ that uniformly randomly drops a token until the sequence is empty. Let $p_\theta$ be the parametric time reversal of the noising process. Then the evidence lower bound for the log-likelihood of the data is given by:

$$\log p_\theta(x_0) \geq \mathbb{E}_{\boldsymbol{x}_{1:T} \sim q_{\boldsymbol{x}_{1:T} \mid \boldsymbol{x}_0}} \left[\log p_\theta(\boldsymbol{x}_0, \boldsymbol{x}_{1:T}) - \log q(\boldsymbol{x}_{1:T} \mid \boldsymbol{x}_0)\right]$$

$$= \mathbb{E}_{\boldsymbol{x}_{1:T} \sim q_{\boldsymbol{x}_{1:T} \mid \boldsymbol{x}_0}} \left[\sum_{t=1}^{T} \log \frac{p_\theta(\boldsymbol{x}_{t-1} \mid \boldsymbol{x}_t)}{q(\boldsymbol{x}_t \mid \boldsymbol{x}_{t-1})} + \log p_\theta(\boldsymbol{x}_T)\right]$$

$$\geq \mathbb{E}_{\boldsymbol{x}_{1:T} \sim q_{\boldsymbol{x}_{1:T} \mid \boldsymbol{x}_0}} \left[\sum_{t=1}^{T} \log p_\theta(\boldsymbol{x}_{t-1} \mid \boldsymbol{x}_t)\right],$$

where in the last step we used the fact that $q$ is fixed and $\log p_\theta(\boldsymbol{x}_T)$ is zero because $\boldsymbol{x}_T$ is always the empty sequence for large enough $T$. Breaking the expression down into a sum over the time steps, we get

$$\mathcal{L}^{\text{mc}}(\theta; \boldsymbol{x}_0) = - \mathbb{E}_{t \sim \mathcal{U}[1,T]} \mathbb{E}_{\boldsymbol{x}_{t-1}, \boldsymbol{x}_t \sim q_{\cdot \mid \boldsymbol{x}_0}} \log p_\theta(\boldsymbol{x}_{t-1} \mid \boldsymbol{x}_t).$$

This loss based on the naive Monte Carlo estimate of the ELBO is easy to compute. However, it is intractable to train a denoising model using this due to two main reasons. First, the estimator can have extremely high variance and therefore unstable to train. Second, parameterizing the denoiser using any standard neural network for sequence modeling like a transformer or LSTM is inefficient because the only one token will be inserted in $\boldsymbol{x}_t$ to obtain $\boldsymbol{x}_{t-1}$, which leads to weak gradients and slow convergence.

We can use the usual trick to utilize $\boldsymbol{x}_0$ to reduce the variance of the estimator (Ho et al., 2020).

$$\log p_\theta(\boldsymbol{x}_0) \geq \mathbb{E}_{\boldsymbol{x}_{2:T} \sim q_{\boldsymbol{x}_{2:T} \mid \boldsymbol{x}_0}} \sum_{t=2}^{T} \mathbb{D}_{\text{KL}}\left[q(\boldsymbol{x}_{t-1} \mid \boldsymbol{x}_t, \boldsymbol{x}_0) \,\|\, p_\theta(\boldsymbol{x}_{t-1} \mid \boldsymbol{x}_t)\right] + \mathbb{D}_{\text{KL}}[q(\boldsymbol{x}_T \mid \boldsymbol{x}_0) \,\|\, p_\theta(\boldsymbol{x}_T)]$$

$$\implies \mathcal{L}^{\text{mc}}(\theta; \boldsymbol{x}_0) = - \mathbb{E}_{t \sim \mathcal{U}[1,T]} \sum_{\boldsymbol{x}_{t-1}} q(\boldsymbol{x}_{t-1} \mid \boldsymbol{x}_t, \boldsymbol{x}_0) \log p_\theta(\boldsymbol{x}_{t-1} \mid \boldsymbol{x}_t).$$

where we make use of the Bayes rule and the Markov assumption to get $q(x_t \,|\, x_{t-1}) = \frac{q(x_{t-1} \,|\, x_t, x_0)\, q(x_t | x_0)}{q(x_{t-1} | x_0)}$ and use it in the expression for ELBO.

When $p_{\text{data}}$ is such that only sequences that do not repeat tokens are in the support of the distribution, then $q(\boldsymbol{x}_{t-1} | \boldsymbol{x}_t, \boldsymbol{x}_0) = d(k, v;\ \boldsymbol{x}_0, \boldsymbol{b})$ where $\boldsymbol{b}$ is such that $\boldsymbol{x}_t = \boldsymbol{x}_0[\boldsymbol{b}]$. Moreover, $p_\theta(\boldsymbol{x}_{t-1} | \boldsymbol{x}_t)$ can be written as $p_\theta(k, v \mid \boldsymbol{x}_t)$. When $p_{\text{data}}$ does not have this property, then we need to use a dynamic programming algorithm to compute all possible alignments of $\boldsymbol{x}_t$ w.r.t $\boldsymbol{x}_0$ to obtain a closed form expression for the loss.

