# OpenReview forum: "Insertion Language Models: Sequence Generation with Arbitrary-Position Insertions"
_ICLR.cc/2026/Conference — Submitted to ICLR 2026_

### Official Review · Reviewer_omE6 · 2025-10-27

**Soundness:** 3
**Presentation:** 4
**Contribution:** 2
**Rating:** 4
**Confidence:** 3

**Summary:**

This paper introduces a method for sequence generation that learns to insert tokens at arbitrary positions in a sequence. The authors refer to this as Insertion Language Models (ILM), and compare it against autoregressive models and masked diffusion models.
ILMs are trained using a single transformer encoder with a simple denoising objective, where tokens are completely dropped from the sequence and the model learns to re-insert them.

On synthetic planning tasks, ILMs are shown to overcome the failure modes of the other models. In particular it performs the best on Star Graphs from (Bachmann & Nagarajan, 2024). For unconditional text generation, ILMs perform competitively with auto regressive models and better than diffusion models.

**Strengths:**

The paper is well written and easy to follow. In particular, I find the figures to be pedagogical and helpful in presenting key ideas and concepts of the paper. I also commend the authors for a thorough and fleshed out related work section and a beautiful appendix. In general, the presentation of the paper comes across as strong, and it’s clear that a lot of effort went into it. (some suggestive further improvements are listed in the Weaknesses section)

In regards to the method, I personally find the proposed insertion method itself to be quite elegant and interesting. The explanation of the experiments and the corresponding results are also clear. The authors also demonstrate strong results on the Star Graphs from (Bachmann & Nagarajan, 2024).

**Weaknesses:**

In general, my concerns regard the contributions of the paper and how they compare to autoregressive modelling(ALM). This is because ALMs are clearly the dominant paradigm, so for a new method to be considered it needs a strong selling point. From what I can see currently, the biggest selling point is that ILM solves the toy problem of Star Graphs from (Bachmann & Nagarajan, 2024).

**Concern 1:**

One could quite easily create a setup where an autoregressive model is capable of performing infilling. Such as existing Fill-in-the-middle setups, or something simple as: If context X is the full text, and X* is the text with removed segments, you could formulate the autoregressive task as: X* -> X.
Perhaps you calculate the loss only on X, or not…

For a more convincing narrative, I would therefore suggest incorporating such a baseline for at least the 5.2.2 Infilling task.

Furthermore, as we scale ALMs and their capabilities and generalizations increase, it is possible that such capabilities would emerge regardless. Of course it’s an unfair comparison in terms of resources, but I’m quite confident that SOTA LLMs would do exceptionally well on these smaller infilling tasks, even if they have not been directly trained towards them. In particular, I’m also confident that these models solve the Star Graphs from (Bachmann & Nagarajan, 2024). (which leads to Concern 2)


**Concern 2:**

Scaling of ILM.

Considering that autoregressive models are so well-established, and one of their strongest properties is how well they scale and generalize, it would be beneficial for you to demonstrate the scaling properties of ILM. Both in terms of computational costs, but also performance.

I suggest to therefore provide metrics for the computational cost for ILM, both in terms of training and inference. And preferably also how these metrics scale with model size, sequence length etc…
Additionally, varying the model/dataset sizes for language modelling, would make for an interesting experiment.

Conveniently, you only explicitly compare the inference time between ILM and MDM. However, in the discussion you mention that inference might be slower for ILM. But how much slower? This seems like a serious limitation if one wishes to scale this to reasoning models, which generate several thousand tokens per answer. In my opinion the applicability of ILM hinges quite a lot on these scaling factors, and might change the overall narrative of your paper.

**-----------------------------------------------------**

**Suggested Presentation Improvements**

Several paragraphs are quite long and dense. It might be more easily digestible for readers to split them up. I understand that this may in part be an attempt to achieve nice formatting. But if possible I would consider tweaking this. Example paragraphs are: Line 33 - 53, Line 291 - 309, Line 334 - 352

Line 17 + Line 54 VS Line 86: can come across as contradicting each other. First you claim to “revisit” insertion based sequence generation. Then Line 86 introduces “Insertion Language Models”. Perhaps make this distinction more clear.

Line: 127: Since the bold text “Limitations of MDMs” start the paragraph, I’m left with the impression that you claim that step size “s” is a limitation of MDMS. But I would argue this is rather a limitation of how people potentially would train MDMs. Similarly, I would find it a bit weird to say that “a high learning rate breaks training for Transformers” and hence this is a limitation…. I therefore suggest some rephrasing here.

Line 208: I have never come across the term “Unembedding” layer, to refer to the final linear layer to the vocab. Perhaps there’s something more common you could pick?

Line 335: Zebra Puzzles are now notated with bold, and reside inside the 5.1.1 Star Graphs sub chapter. Seems more appropriate and consistent to let Zebra Puzzles be its own subchapter.

**Questions:**

See weaknesses for concerns

---

> ### Author Response · Authors · 2025-11-21
> **Thank you for your feedback and suggestions**
>
> Thank you for appreciating the clarity and the presentation of our paper. Below, we address the two concerns you mention in your review.
>
> > Concern 1: Infilling using ARMs
>
> Large ARMs are quite capable and could be specially fine-tuned to perform restricted forms of infilling like fill-in-the-middle, where there is a single chunk of text that is missing. However, fill-in-the-middle is not a standard approach---it does not have an accepted standard open-source code---and therefore is not a reasonable *baseline* to implement and compare fairly within the scope of this paper. Please consider the questions one would need to answer to implement it in our setting: What should be the data mix of fill-in-the-middle and normal ARM training on our corpora? Should there be a curriculum? How do we handle the case of multiple missing chunks? We are not suggesting that ARMs are incapable of limited forms of infilling, but training such a model in itself is a significant undertaking and is not a reasonable baseline within the scope of a our paper.
>
> > provide metrics for the computational cost for ILM, both in terms of training and inference
>
> **Training**: At the scale of small (GPT-2 sized) language models, the training for ILMs is as efficient as ARMs and MDMs. To show this, we present the mean and standard deviation of iteration times for training on padded sequences.
> These are iteration times (iterations per second) for training the gpt2-small scale model (~85M non-embedding params, ~170M total params) on 4 A100 GPUs, each on its own node (i.e., one GPU per node DDP setup) with per-device per-iteration batch size of 64, with `torch.compile` and `bfloat16` precision.
> | Model | Dataset | seq. len. | mean itr/sec | std itr/sec |
> |-------|---------|-----------|-------------|-------------|
> | ILM   | Stories    | 1024      | 1.73         | 0.14         |
> | ARM   | Stories    | 1024      | 1.71         | 0.11         |
> | MDM   | Stories    | 1024      | 1.72         | 0.12         |
>
> As you can see, on padded sequences, there is a negligible difference in training efficiency between ILMs, ARMs, and MDMs. This is due to the fact that we perform noising in the collator using multiple workers so the only work done on the GPU is the forward pass, loss computation with precomputed targets, and the backward pass. All of these have almost identical computational costs for all three models.
>
> **Inference**: The inference cost of ILMs is identical to that for ARMs without KV caching.
> We have added ARM inference time, with and without KV caching to Figure 6 in the revised draft.
>
> > how do metrics scale with model size, sequence length etc…?
>
> We include experiments on two corpora LM1B and Stories, both of which have very different characteristics. LM1B, a corpus based on the news domain, has a large vocabulary and contains short sequences up to 128 tokens. Stories corpus, on the other hand, has a smaller vocabulary, but is more open-ended, and contains up to 1024 tokens.
> The goal of this paper is to highlight the key advantages of ILMs at a small scale.
> Training larger models on larger corpora would be relevant, but performing a thorough, systematic scaling study by training models at many different scales is highly resource-intensive and beyond the scope of this paper. It would definitely be an interesting avenue for follow-up work if sufficient compute is available.
>
> We do a small ablation study on the star graph task to understand the impact of model size on the synthetic tasks. As shown in the table below, we find that ILM largely maintains almost perfect accuracy even for smaller-sized models, except for the smallest size, where its per-token accuracy drops below that of ARM and MDM.
>
> | Model | size | easy (token acc) | easy (seq. acc) | medium (token acc) | medium (seq. acc) | hard (token acc) | hard (seq. acc) |
> | ----- | ---- | ---------------- | --------------- | ------------------ | ----------------- | ---------------- | --------------- |
> | ILM | 85M | 100.0 | 100.0 | 100.0 | 100.0 | 99.1 | 99.7 |
> | ARM | 85M | 81.7 | 32.3 | 81.7 | 75.0 | 43.2 | 23.0 |
> | MDM | 85M | 100.0 | 100.0 | 90.6 | 21.0 | 54.9 | 21.0 |
> | | | | | | | | |
> | ILM | 42M | 100.0 | 100.0 | 100.0 | 100.0 | 97.1 | 95.7 |
> | ARM | 42M | 55.1 | 32.13 | 91.8 | 89.0 | 33.9 | 14.1 |
> | MDM | 42M | 99.9 | 99.5 | 92.2 | 42.9 | 57.8 | 0.54 |
> | | | | | | | | |
> | ILM | 8M | 100.0 | 100.0 | 100.0 | 100.0 | 12.3 | 2.3 |
> | ARM | 8M | 55.4 | 32.7 | 73.0 | 60.4 | 22.1 | 1.4 |
> | MDM | 8M | 48.1 | 0.5 | 68.3 | 12.1 | 35.5 | 0.0 |
>
>
> **Regarding Presentation**:
>
> Thank for your thoughtful suggestions. We will incorporate these into the revised draft.
>
> ---
>
> We are happy to answer any follow-up questions that you may have.

---

> ### Author Response · Authors · 2025-11-27
> **Follow up**
>
> As the discussion period is coming to a close, we would like to kindly follow up. We have taken great care to address your questions and concerns, and if you have any further questions or comments regarding our work, we would be more than happy to discuss and respond before the deadline.
>
> Thank you again for your time and valuable feedback!

---

### Official Review · Reviewer_Tp3B · 2025-11-01

**Soundness:** 4
**Presentation:** 4
**Contribution:** 3
**Rating:** 6
**Confidence:** 4

**Summary:**

The paper introduces ILMs, a novel sequence generation paradigm that predicts both token and insertion position jointly. This idea generalizes ARMs and MDMs by allowing arbitrary generation order and variable-length outputs, elegantly overcoming key limitations of both frameworks. The authors demonstrate the performance improvements and increased flexibility experimentally on various tasks.

**Strengths:**

1. The authors clearly articulate the shortcomings of existing ARMs (sequential bias) and MDMs (fixed-length masks, simultaneous unmasking). ILMs offer a different approach with many benefits, and the authors' presentation of this is clear.

2. The experiments are well chosen and convincing. The paper evaluates ILMs on synthetic planning tasks (star graphs, zebra puzzles) and on realistic text datasets (LM1B, TinyStories). ILMs outperform ARMs and MDMs on constrained reasoning tasks and perform competitively on text generation, while uniquely supporting arbitrary-length infilling. I think it is good when an experiment offers conclusive answers, since it allows the field to build on solid ground. With that in mind, I feel that these experiments convey the relative improvements on the tasks considered.

**Weaknesses:**

1. Not allowing caching is a pretty big deal for incremental generative models. I understand that it is future work, but it could be nice to discuss a bit what the possible avenues are toward efficient ILM inference?

2. I always feel bad saying this, but it would be nice to see how results change at larger scales. I know this is easier said than done, and scaling up experiments is usually the goal anyway. However, my point is that sometimes the extra flexibility/expressivity becomes harder for the model to make use of at larger scales (like needle in a high-dim haystack), and sometimes it becomes easier (it enables a realistic, generalizable solution instead of some impractical and unrobust best fit). Everyone has their thresholds somewhere, but for myself (and many people I've worked with) it's hard to get any signal on this with sequence lengths and complexity as restricted as in e.g. LM1B. To summarize, this isn't a weakness so much as something to look out for, but it would be nice to see some experiments/exposition put toward the question "what parts of ILMs do we expect to become more robust/more brittle at scale"?

**Questions:**

1. The experiments discuss the per-token generation time, NLL, etc., but I would like to know more about the qualitative ways in which ILM generations differ from ARM/MDM generations at larger scales. Are the sequences more complicated/higher entropy/longer? Is there some patter/trend for how models tend to use insertions that the authors anecdotally saw?

2. Any ideas how to address caching/fast inference? There is also growing interest in pre-compiled deep learning models (jax.jit, torch.compile, ...) and I imagine ILMs would face all kinds of issues about not knowing lengths of arrays in advance for preallocation. Are the authors looking toward efficient implementation?

---

> ### Author Response · Authors · 2025-11-21
> **Efficiency and KV caching**
>
> Thank you for highlighting the clarity and the contribution of our work. We appreciate the motivation for your questions and concerns. We will first answer your questions regarding efficiency, and discuss qualitative observations and scaling in a separate response.
>
> > how to address caching/fast inference?
>
> One cannot maintain a KV cache for ILMs in the same way as ARMs is because as the tokens are inserted, the positions of the old tokens keep getting shifted.
> One way to fix this is to construct "state" such that the position information is decomposed into *relative* and *absolute*, where the relative positions of two tokens in the sequence never changes.
> For example, consider a sequence representation that stores the list of tokens $w_1, w_2, \ldots, w_n$ that can be out of order and a matrix $r_{ij}$, where
> $$\begin{align}r_{ij} = \begin{cases}
>     1 & \text{if } \text{pos}(w_j) > \text{pos}(w_i);~~~~ \\
>     -1 & \text{if } \text{pos}(w_i) > \text{pos}(w_j); ~~~~~\\
>     0 & \text{if } \text{pos}(w_i) = \text{pos}(w_j).
> \end{cases}\end{align}$$
> When a new token is inserted between $w_i$ and $w_j$, we can append it to the list and we add a row and column to the matrix $r_{ij}$ which will store the new tokens relative position [1]. Now this state can be cached just like ARMs. But we need some additional layers on top that can inject the current relative distance information into the final representations. This would be an exact KV caching mechanism, which would only be slightly more expensive than ARMs.
>
> Alternatively, one can force the insertion training objective to have semi-autoregressive insertions, meaning that insertions are limited only to a rolling block of positions that move rightwards. We would lose some of the advantages of arbitrary generation order, but we can perform approximate delayed KV caching [2,3].
>
> > compiled deep learning models (jax.jit, torch.compile,)...issues about not knowing lengths of arrays in advance for preallocation
>
> We pad sequences to max length so the arrays can be pre-allocated. Our ILM training and inference loop can be compiled without any graph breaks using `torch.compile` in torch 2.5.0 and later. All our models are trained with `torch.compile` and in `bfloat16` precision. At the scale of small (GPT-2 sized) language models, the training for ILMs is as efficient as ARMs and MDMs. To show this, we present mean and std of iterations times for training on padded sequences.
> These are iteration times (iterations per second) for training gpt2-small scale model (~85M non-embedding params, ~170M total params) on 4 A100 GPUs, each on its own node (i.e., one GPU per node DDP setup) with per-device per-iteration batch size of 64, with `torch.compile` and `bfloat16` precision.
> | Model | Dataset | seq. len. | mean itr/sec | std itr/sec |
> |-------|---------|-----------|-------------|-------------|
> | ILM   | Stories    | 1024      | 1.73         | 0.14         |
> | ARM   | Stories    | 1024      | 1.71         | 0.11         |
> | MDM   | Stories    | 1024      | 1.72         | 0.12         |
>
> As you can see, on padded sequences, there is negligible difference in training efficiency between ILMs, ARMs, and MDMs. This is due to the fact that we perform noising in the collator using multiple workers so the only work done on the GPU is the forward pass, loss computation with precomputed targets, and the backward pass, all of these have almost identical computational cost for all three models.
> Additionally, we have also looked into FlexAttention [8], to gain the ability to pack sequences, and to match the efficiency of FlashAttention (which does not support padding). Our current implementation does not use FlexAttention, but as the support for FlexAttention improves in Pytorch, we the pre-training of ILMs can also be performed with packed sequences achieving parity with ARMs and MDMs in terms of training efficiency.
>
> ### References
> 1. Gu, Jiatao, Qi Liu, and Kyunghyun Cho. "Insertion-based decoding with automatically inferred generation order." Transactions of the Association for Computational Linguistics 7 (2019): 661-676.
> 2.  Ma, Xinyin, et al. "dkv-cache: The cache for diffusion language models." arXiv preprint arXiv:2505.15781 (2025).
> 3. Arriola, Marianne, et al. "Block diffusion: Interpolating between autoregressive and diffusion language models." arXiv preprint arXiv:2503.09573 (2025).
> 4. Vogels, Thijs, Sai Praneeth Karimireddy, and Martin Jaggi. "PowerSGD: Practical low-rank gradient compression for distributed optimization." Advances in Neural Information Processing Systems 32 (2019).
> 5. Nie, Shen, et al. "Large language diffusion models." arXiv preprint arXiv:2502.09992 (2025).
> ---
>
> We address your questions on scaling and qualitative analysis in the next response.

---

> ### Author Response · Authors · 2025-11-21
> **Regarding qualitative observations**
>
> > qualitative ways in which ILM generations differ from ARM/MDM generations
>
> In general, we observe that the ILM's generations are shorter than MDM's, which is one of the reasons for higher coherence scores for ILM. MDM due to its longer generations, messes up entities and long range consistency more frequently.
> Repetition manifests in ILM's generations differently due to its ability to insert. Specifically, we observe that there is a pattern that generations have some alliterations, like short phrases "he put", "they run", etc. appear at many locations in the same generation.
> We believe that this is due the token frequencies in the Stories data set. These simple phrases are quite frequent in the dataset and therefore ILM inserts many of them early in the generation process and then fills in more nuanced tokens.
> We also find that ILM's generations have lesser grammatical mistakes than MDM as shown in our LLM judge evaluation in Figure 5.
> To support these observations we provide comparative examples in Appendix B.0.6 in our original submission. Additionally, we have also added this exposition and complete generation trajectories from ILM in Appendix C.0.4.
>
>
> > it would be nice to see some experiments/exposition put toward the question "what parts of ILMs do we expect to become more robust/more brittle at scale"?
>
>
> We understand the intention behind this question.
> In order to have a fair comparison, we need to train all three models on larger corpus (eg. OpenWebText) preprocessed to have variable length sequences. Due to limited compute, we are unable to train all three types of models on a large corpus. However, we already make some observations based on our current experiments on LM1B and Stories, which have different characteristics. LM1B, a corpus based on news domain, has a large vocabulary, and contains short sequences up to 128 tokens. Stories, on the other hand, has a smaller vocabulary, but is more open ended, and contains up to 1024 tokens.
>
> 1. One notable observation, which we also discuss above, is ILM's preference for shorter seqeunces in case of open ended story generations.
> 2. As shown by the generation trajectories in Appendix C.0.4 of the revised draft, the ILM's generations are highly non-autoregressive due to the models perception of sequences in relative positions. On the other hand, due to its perception of positions in absolute terms, it has been observed that  confidence based generation for MDMs shows preference for semi-autoregressive order at large scale [5], which limits the number of tokens MDMs can generate per step. Future work can explore, multi-token generation for ILMs at a larger scale as we discuss in our response to reviewer Mg39.
>
> ### References
> 1. Gu, Jiatao, Qi Liu, and Kyunghyun Cho. "Insertion-based decoding with automatically inferred generation order." Transactions of the Association for Computational Linguistics 7 (2019): 661-676.
> 2.  Ma, Xinyin, et al. "dkv-cache: The cache for diffusion language models." arXiv preprint arXiv:2505.15781 (2025).
> 3. Arriola, Marianne, et al. "Block diffusion: Interpolating between autoregressive and diffusion language models." arXiv preprint arXiv:2503.09573 (2025).
> 4. Vogels, Thijs, Sai Praneeth Karimireddy, and Martin Jaggi. "PowerSGD: Practical low-rank gradient compression for distributed optimization." Advances in Neural Information Processing Systems 32 (2019).
> 5. Nie, Shen, et al. "Large language diffusion models." arXiv preprint arXiv:2502.09992 (2025).
>
> ---
>
> We hope that our response addresses your concerns. We are happy to answer any follow-up questions you may have.

---

> ### Author Response · Authors · 2025-11-27
> **Follow up**
>
> As the discussion period is coming to a close, we would like to kindly follow up. We have taken great care to address your questions and concerns, and if you have any further questions or comments regarding our work, we would be more than happy to discuss and respond before the deadline.
>
> Thank you again for your time and valuable feedback!

---

> > ### Comment · Reviewer_Tp3B · 2025-11-28
> >
> > I thank the authors for their clear and detailed responses. I appreciate the details regarding caching and fast inference, which reflect the amount of engineering effort and careful thought the authors have devoted to practical considerations. My question regarding qualitative properties of the ILM generations is admittedly a strange and loose one: the authors' response does a good job of conveying the qualitative observations which are spread out throughout the main paper, and I thank them for that.

---

### Official Review · Reviewer_Mg39 · 2025-11-01

**Soundness:** 3
**Presentation:** 3
**Contribution:** 2
**Rating:** 6
**Confidence:** 3

**Summary:**

The authors revisit Insertion Language Models (ILMs), an insertion-based language model for arbitrary-length, arbitrary-order token generation. This model predicts output tokens along with their positions in the existing sequence, one-at-a-time.

The key contributions are:
1. Addressing the high variance problem of naive infilling denoising objectives through an approximate denoising training objective and tailored parameterization of the denoising network.
2. Demonstrating that ILMs outperform ARMs and MDMs on synthetic tasks (path generation on star graphs, zebra puzzles).
3. Showing that ILMs perform slightly better than MDMs and are competitive with ARMs on language generation tasks.

**Strengths:**

The paper successfully modernizes the classical idea of insertion-based language models by combining it with denoising objectives. The performance improvements are convincingly demonstrated on synthetic tasks, clearly showing the advantages of the proposed approach.

As a reference, it may be helpful to also cite:
- Insertion-based Decoding with automatically Inferred Generation Order, https://arxiv.org/abs/1902.01370

**Weaknesses:**

- **Method: Fundamental Inefficiency**

The primary limitation of ILMs is that they sacrifice parallelization benefits from both ARMs and MDMs. Unlike ARMs, ILMs cannot leverage efficient parallel training, and unlike MDMs, they do not support parallel inference. However, despite giving up these parallelization advantages, the improvement in generative perplexity is not substantial (Table 2, LM1B).

- **Experiment Setup: NLL Measurement**

NLL measures how accurately a model captures the learned distribution. However, the comparison appears unfair: MDM uses naive tau-leaping sampling while ARM and ILM use nucleus sampling with p=0.9. For a fair comparison, I recommend re-evaluating with p=1.0 for all models.

- **Experiment Setup: Infilling Task**

Similar to the NLL experiments, either: (1) change p=0.9 to p=1.0 for fair comparison, or (2) employ advanced sampling strategies for MDM (e.g., confidence-based sampling methods [1]) to verify performance under more favorable conditions.

- **Experiment Setup: Baselines**

Recent discrete diffusion models such as Edit Flows [2] and FlexMDM support arbitrary-length generation. However, comparisons with Edit Flows are missing from the synthetic task experiments. I would like to see: (1) performance comparisons between ILMs and Edit Flows, and (2) clarification of what advantages insertion-based language models offer over editing-capable discrete diffusion/flow matching models.

**References:**
- [1] Train for the Worst, Plan for the Best: Understanding Token Ordering in Masked Diffusions, https://arxiv.org/abs/2502.06768
- [2] Edit Flows: Flow Matching with Edit Operations, https://arxiv.org/abs/2506.09018

**Questions:**

- (Minor) Why wasn't the per-token generation time vs. NLL figure presented as the more conventional NFE vs. generative perplexity plot? Also, could you please include ARM in this comparison?

---

> ### Author Response · Authors · 2025-11-21
> **Efficiency and Sampling hyperparameters**
>
> Thank you for highlighting the clarity and the contributions of our work, and for the insightful questions. Below, we address each of your questions about inference efficiency and MDM sampling parameters. In the subsequent response, we provide a comparison with EditFlow and FlexMDM.
>
> > ..unlike MDMs they (ILMs) do not support parallel inference
>
> We believe that the slow inference for ILM is not a fundamental issue, and can be addressed in follow-up work.
> The stopping classifier based formulation can be generalized to allow multiple insertions as the token probabilities are already available at multiple gaps. To show that the apparent limitation is not fundamental, we perform unconditional generation using multiple insertions using a fixed schedule, where we select top-r gaps using the marginal  $\{k_1, \dots, k_r\}=top_{r} \sum_{v} p_{ilm}(k, v | x[b])$  and then sample the insertions using $v_i \sim p_{ilm}(v |x[b], k_i)$. We use a fixed schedule of first 10% steps during the generation to perform multiple insertions, followed by single token insertions. Following are the results for $r=2,3$ (the results for the default $r=1$, which are presented in the main paper, are presented in bold). We find that there is no significant degradation in the NLL even if we perform multiple insertions.
>
> | dataset | r | NLL | Ent. |
> |---------|-----|-----|-----|
> | **LM1B** | 1 | **4.67** | **2.80** |
> | LM1B | 2 | 4.71 | 2.79 |
> | LM1B | 3 | 4.85 | 2.82 |
> | **Stories** | 1 | **2.14** | **3.76** |
> | Stories | 2 | 2.12 | 3.72 |
> | Stories | 3 | 2.33 | 3.68 |
>
>
> > Sampling parameters for MDM
>
> The results presented for MDMs use tau-leaping but with max_steps=1024, i.e., an extremely low $\Delta t$, which effectively produces less than one token per forward pass. Additionally, once the sampler samples a position to unmask, we use nucleus value p=0.9 for the token selection, just like we do for ARMs.
> We also tried using greedy token selection based on top-2 probability difference as suggested in [1], but it performed worse than random token selection. Upon closer inspection, we find that for short padded sequences like LM1B and TinyStories, the greedy selection is susceptible to early sampling of PAD tokens (also observed in [4], even for large MDMs trained with PAD tokens), and also high token repetitions.
> We find that the sampling hyperparameters used to report our current MDM results produce the best results on LM1B and Stories for the model.
> We present the results of greedy position selection on LM1B below. The bold row corresponds to the settings and results presented in our original submission.
>
>
> | Dataset | Model | Pos. Selection | token selection p| NLL | Ent. | Avg. Len. |
> |---------|---------------|-----------------|-----|-----|-----|-----|
> | **LM1B**    | **MDM**   | **random**| **0.9** | **4.81** |  **3.70**| **85** |
> | LM1B    | MDM   | greedy [1]| 0.9 | 5.11 |  2.50| 63 |
> | **LM1B**    | **ILM**   | **ancestral greedy** | **0.9** | **4.67** | **2.80** | **21** |
> | LM1B    | ILM   | None | 1.0 | 4.18 | 2.82 | 22 |
> | **Stories** | **MDM**   | **random**| **0.9** | **2.54** |  **4.55** | **985** |
> | Stories | MDM   | greedy [1]| 0.9 | 2.33 |  4.18| 885 |
> | **Stories** | **ILM**   | **ancestral greedy** | **0.9** | **2.14** | **3.76** | **119** |
> | Stories | ILM   | None | 1.0 | 2.15 | 3.72 | 113 |
>
>
>
> We also find that for ILM using ancestral sampling, i.e., picking a token after the position, does not provide a big advantage.
> In the revised draft, we have added the plot (figure 9 in Appendix C) of NLL vs Entropy for ILM for different sampling hyperparameters. We will incorporate a discussion around the sampling hyperparameters in the main text.
>
> > NLL vs Time plot
>
> In Figure 6, we present the x-axis in terms of time (instead of number of function evals) to highlight the fact that each forward pass of MDM costs the same, but that is not the case for the ILM and the ARM.
> In the revised draft, we have also added ARM inference time, with and without KV caching, to the plot.
>
> In the following response, we discuss EditFlow and FlexMDM and their relation to our work.
>
> ### References
> 1. Train for the Worst, Plan for the Best: Understanding Token Ordering in Masked Diffusions
> 2. Havasi, Marton, et al. "Edit Flows: Flow Matching with Edit Operations." arXiv preprint arXiv:2506.09018 (2025).
> 3. Kim, Jaeyeon, et al. "Any-Order Flexible Length Masked Diffusion." arXiv preprint arXiv:2509.01025 (2025).
> 4. Kim, Bumjun, et al. "Rainbow Padding: Mitigating Early Termination in Instruction-Tuned Diffusion LLMs." arXiv preprint arXiv:2510.03680 (2025).

---

> ### Author Response · Authors · 2025-11-21
> **Comparison with Edit Flows and FlexMDM**
>
> We cite FlexMDM [3] and Edit Flows [2] in our extended related work as concurrent works, and we appreciate your request for a discussion and comparison.
> However, at the time of submission both works were available solely on ArXiv, published within 2 months of the ICLR submission deadline, and with no links to opensouced code. We hope that you understand that as acknowledged in ICLR's policy on concurrent work (https://iclr.cc/Conferences/2026/ReviewerGuide, last FAQ), it is very difficult for us to have an empirical comparison with these at the time of submission.
>
> That said, we make the best effort to provide an empirical and conceptual comparison with both FlexMDM and Edit Flows within the available time frame.
>
> **Conceptual comparison:**
>
> **ILMs are time-agnostic InsertionFlows**
>
> We can show that the EditFlow loss in equation (23) in [2], when viewed as an insertion-only objective, decomposes into a stopping loss and a token selection loss, conceptually equivalent to our formulation in section 3. Specifically, the loss in equation (23) in [2], if insertion is the only operation considered, decomposes as (without the outer expectation):
>
> $$
> \begin{align}
> = \sum_{i=0}^{\text{len}(x_t)} \lambda_{t, i}^{\text{ins}}(x_t) - \frac{\dot\kappa_t}{1 - \kappa_t} \Delta s_t[i]\log \lambda_{t, i}^{\text{ins}}(x_t)\\
> \quad - \quad\frac{\dot\kappa_t}{1 - \kappa_t}\sum_{i=1}^{\text{len}(x_t)}\sum_{j\in s_t[i+1]-s_t[i]} \log Q_{t,j}^{\text{ins}}(z_t^j|x_t)\\
> \end{align}
> $$
> Inserting the data expectation outside, we see that the first term is minimized when $\lambda_{t, i}^{\text{ins}*}(x_t) = \frac{\dot\kappa_t}{1 - \kappa_t}\mathbb E(\Delta s_t[i])$.
> For a small $\Delta t$, the predicted probability of stopping all insertions at the current sequence $x_t$ is $\mathbb P(\text{stop}) = 1 - \prod_{i=1}^{\text{len}(x_t)}\lambda_{t, i}^{\text{ins}}(x_t) {\Delta t}$. With the optimal minimizer, we see that $\mathbb P(\text{stop})$ goes to 1 when $\mathbb E(\Delta s_t[i]) \to 0$, i.e., all predicted gap lengths are 0. This is precisely what our stopping classifier learns to predict.
> Finally, the second loss term is a differently weighted (time dependent weight) version of our token selection loss.
>
> We can perform a similar analysis for FlexMDM to observe that a time-agnostic version of FlexMDM will be equivalent to a *per-gap* stopping classifier, exactly the same as above, plus a token selector. The key differences are that the token selection in FlexMDM takes in a mask tokens as input as opposed to a single gap, and the stopping classifier is replaced by regressors that predict length of each gap, which is then converted into the Poisson rate of insertions.
>
> **Empirical comparison:**
>
> We implement both FlexMDM and EditFlow in our codebase. Due to limited time, we are only able to provide results on the synthetic task and LM1B. In the table below (T) and (E) denote the token and exact sequence match, respectively.
>
> **Star Graphs:** For FlexMDM, we use the confidence-based decoding strategy proposed in [3], and for EditFlow, we train the *default EditFlow* as described in Section 5 in [2], which does not use substitutions, just insertions and deletions. In an effort to select the most favorable sampling hyperparameters, for both models we use tau-leaping to step forward time but with max_step=1024, which corresponds to an extremely low $\Delta t$, effectively producing less than 1 token per forward pass on average.
>
> | Model | Easy (T) | Easy (E) |Medium (T) | Medium (E) |  Hard (T)| Hard (E)|
> |-------|-----------|-------------|-----------|-----------|-----------|-----------|
> | FlexMDM |91.1 | 68.7 | 89.1 | 94.5 |52.5 | 21.1 |
> | EditFlow |93.3 | 79.9 | 95.3 | 83.5 | 89.7 | 67.1 |
>
> Both FlexMDM and EditFlow perform better than MDM (Table 1 in the paper).
> For both FlexMDM and EditFlow, we observe that most sampling trajectories contain the correct subsequence of the target sequence till some point; however, towards the end of generation, the model overshoots the target. This is due to the presence of one regression output per token for both these models, which parameterizes the respective Poisson insertion rates. For tasks that have exactly one correct response, having a single stopping classifier is advantageous.
>
> **LM1B**
> On LM1B, for FlexMDM, we do not use confidence-based position selection, because we observe similar issues as with MDMs on short sequences, i.e., excessive token repetitions (very low entropy around 1). We present the result for FlexMDM trained on LM1B in the table below.
>
> | Model | NLL | Ent. | Avg. Len. |
> |-------|-----|-----|-----|
> | FlexMDM | 5.26 | 3.19 | 33.4 |
>
>
> We observe that most samples from FlexMDM are of good quality; however, some have token repetitions, bringing down the overall average NLL.
>
> ---
>
> We hope that through our detailed responses, we were able to address most of your concerns. We will incorporate the same in the paper. Happy to answer any follow-up questions that you may have.

---

> ### Author Response · Authors · 2025-11-27
> **Follow up**
>
> As the discussion period is coming to a close, we would like to kindly follow up. We have taken great care to address your questions and concerns, and if you have any further questions or comments regarding our work, we would be more than happy to discuss and respond before the deadline.
>
> Thank you again for your time and valuable feedback!

---

### Official Review · Reviewer_2Tam · 2025-11-04

**Soundness:** 3
**Presentation:** 3
**Contribution:** 2
**Rating:** 4
**Confidence:** 3

**Summary:**

The paper proposes a language modeling framework that enables sequence generation of arbitrary length by extending the existing idea of generation through insertion. The proposed models are evaluated on both planning tasks and language modeling tasks, including unconditional text generation and text infilling. Experimental results demonstrate performance improvements over baseline models such as Autoregressive Models (ARMs) and Masked Diffusion Models (MDMs).

**Strengths:**

The paper makes an original contribution by adapting insertion-based sequence generation to general language modeling, allowing models to generate sequences of arbitrary length. This is a practical extension of existing techniques, expanding their applicability beyond fixed-length generation tasks.

The work is of clear presentation and well-structured experiments on both synthetic and real-world datasets. Results show consistent improvements over strong baselines such as Autoregressive and Masked Diffusion Models. Overall, the paper is clearly written, methodologically sound, and makes a meaningful contribution to advancing flexible and general-purpose language generation.

**Weaknesses:**

The paper’s technical novelty is somewhat limited, as its main contribution lies in applying an existing insertion-based sequence generation technique to the problem of variable-length text generation. While this adaptation is practical, the paper does not sufficiently deepen the theoretical or conceptual understanding of insertion-based generation, nor does it clearly articulate the unique challenges encountered when extending this approach to general language modeling. A more thorough analysis of these challenges, such as handling coherence, positional dependencies, or long-range consistency, would strengthen the contribution.

Furthermore, several (not all) recent studies that have explored similar directions [1,2] appear to have been overlooked. Incorporating a discussion of these works and clarifying how the proposed model differs or improves upon them would better situate the paper within the existing literature and highlight its distinct value. Overall, expanding the theoretical motivation and engaging more critically with prior research would enhance both the novelty and the impact of the work.

[1] Li, J., Dong, X., Zang, Y., Cao, Y., Wang, J., & Lin, D. (2025). Beyond fixed: Variable-length denoising for diffusion large language models. arXiv e-prints, arXiv-2508.
[2] Gu, Y., Wang, W., Feng, X., Zhong, W., Zhu, K., Huang, L.,  & Qin, B. (2024). Length controlled generation for black-box LLMs. arXiv preprint arXiv:2412.14656.

**Questions:**

Could the authors discuss the current state-of-the-art approaches to variable-length text generation and clarify how their method compares to these existing techniques?

---

> ### Author Response · Authors · 2025-11-21
> **Thank you for your feedback**
>
> Thank you for acknowledging the originality, clarity, and methodological soundness as key strengths of our work.
> In our response below, we address your concerns about the technical novelty and the unique challenges of insertion-based language modeling. In a subsequent response, we answer your question about concurrent related work.
>
> > The paper’s technical novelty is somewhat limited...applying an existing insertion-based ... to variable-length text generation.
>
> > does not articulate the unique challenges encountered when extending this approach to general language modeling.
>
> We discuss the technical novelty of our work in the related work section, and the extended related work in Appendix A. However, we appreciate the opportunity to clarify further how our method differs meaningfully from prior insertion-based approaches, and how it addresses some of the unique challenges of language model pre-training. Below, we briefly summarize the discussion from our related work section, focusing on key works.
>
> - **NMTG** [3]: Uses a high-variance RL-based “learning to search” objective that is comparatively much slower to train and constrains inference to level-order binary tree traversal rather than arbitrary insertion order. These limitations make NMTG difficult to scale to larger language modeling corpora.
> Levenshtein Transformer [4] also faces similar scalability challenges, as it uses a dual-policy training objective that does not scale to decoder-only language model pre-training.
> On the other hand, our training approach is as scalable as that of MDMs and ARMs pre-training. Specifically, each gradient step for ILM costs the same as it would for ARMs and MDMs. The training batch, consisting of both inputs and targets, can be prepared in advance on the CPU, as is typically done for ARMs. The loss is a combination of two simple cross-entropy terms, one for token prediction and one for stopping classification, and overall maintains the same training efficiency as ARMs.
>
> - **Insertion Transformer (IT)** [6]: The most comparable prior work, which we directly evaluate against. The IT lacks a specialized stopping classifier and instead predicts a special EOS token from all slots to decide when to stop generation. We empirically demonstrate that this approach is unreliable for variable-length generation, often overshooting or undershooting the target sequence (please see Appendix C.0.2, “Comparison with Insertion Transformer,” for more details). ILM addresses this issue by introducing a specialized stopping classifier.
>
>
>
> > analysis of these challenges, such as handling coherence, positional dependencies, or long-range consistency
>
> Due to the specific nature of language modeling, we evaluate coherence, consistency, fluency, grammaticality, non-redundancy, and spelling accuracy using an open-source LLM judge trained specifically for this purpose. The evaluation prompts are provided in Table 6 of the Appendix. We also provide high as well as low-scoring generation examples in Appendix B. Below, we discuss some generation observations.
>
> In general, we observe that ILM’s generations are shorter than MDM’s, which is one of the reasons for ILM’s higher coherence scores. MDM, due to its longer generations, messes up entities and long-range consistency more frequently.
> Repetition manifests in ILM’s generations differently due to its ability to insert. Specifically, we observe that generations exhibit a pattern: short phrases like “he put”, “they run”, etc. appear at many locations within the same generation.
> We believe that this is due the token frequencies in the Stories data set. These simple phrases are pretty frequent in the dataset, and therefore, ILM inserts many of them early in the generation process and then fills in more nuanced tokens.
> We also find that ILM’s generations have fewer grammatical mistakes than MDM, as shown in our LLM judge evaluation in Figure 5.
> To support these observations, we provide comparative examples in Appendix B.0.6 in our original submission. Additionally, we have also added this exposition and complete generation trajectories from ILM in Appendix C.0.4.
>
>
>
>
>
>
> ### References
>
> [1] Li, J., Dong, X., Zang, Y., Cao, Y., Wang, J., & Lin, D. (2025). Beyond fixed: Variable-length denoising for diffusion large language models. arXiv e-prints, arXiv-2508.
>
> [2] Gu, Y., Wang, W., Feng, X., Zhong, W., Zhu, K., Huang, L., & Qin, B. (2024). Length controlled generation for black-box LLMs. arXiv preprint arXiv:2412.14656.
>
> [3] Welleck et al. "Non-Monotonic Sequential Text Generation." ICML 2019.
>
> [4] Gu et al. "Levenshtein Transformer." NeurIPS 2019.
>
> [6] Stern et al. "Insertion Transformer: Flexible Sequence Generation via Insertion Operations." ICML 2019.

---

> ### Author Response · Authors · 2025-11-21
> **Concurrent related work**
>
> > Could the authors discuss the current state-of-the-art approaches to variable-length text generation and clarify how their method compares to these existing techniques?
>
> We would like to clarify the difference between *length controlled generation* [2] and our work. Length controlled generation aims to generate sequences within a length range given at inference time. This is not the goal of our work. ARMs and MDMs are the two most widely adopted language modeling paradigms, and we compare with both of these approaches in our paper. Therefore, we believe we are comparing with what one might call state-of-the-art sequence generation models, but at a smaller scale.
>
>
> > several (not all) recent studies that have explored similar directions [1,2] appear to have been overlooked
>
> Thank you for bringing [1] to our attention. In our work, we highlight the advantages of the proposed insertion-based generation method over MDMs and ARMs. MDMs struggle with variable-length generation and infilling, and ILMs naturally address this. [1] proposes a method to address the challenge within the framework of MDMs, by inserting [MASK] tokens. ILMs, on the other hand, insert real tokens directly.
> We will add a discussion around it in the related work section in the next revision. However, our work is not directly comparable to [1] because the models used in our work are much smaller and trained from scratch.
>
>
> ### References
>
> [1] Li, J., Dong, X., Zang, Y., Cao, Y., Wang, J., & Lin, D. (2025). Beyond fixed: Variable-length denoising for diffusion large language models. arXiv e-prints, arXiv-2508.
>
> [2] Gu, Y., Wang, W., Feng, X., Zhong, W., Zhu, K., Huang, L., & Qin, B. (2024). Length controlled generation for black-box LLMs. arXiv preprint arXiv:2412.14656.
>
> [3] Welleck et al. "Non-Monotonic Sequential Text Generation." ICML 2019.
>
> [4] Gu et al. "Levenshtein Transformer." NeurIPS 2019.
>
> [6] Stern et al. "Insertion Transformer: Flexible Sequence Generation via Insertion Operations." ICML 2019.
>
> ---
> We believe that we have addressed all of your concerns. We are happy to answer any follow-up questions that you may have.

---

> ### Author Response · Authors · 2025-11-27
> **Follow up**
>
> As the discussion period is coming to a close, we would like to kindly follow up. We have taken great care to address your questions and concerns, and if you have any further questions or comments regarding our work, we would be more than happy to discuss and respond before the deadline.
>
> Thank you again for your time and valuable feedback!

---

### Meta-Review · Area_Chair_TKnZ · 2026-01-05

**Summary:**

This paper proposes a language modeling framework that enables sequence generation of arbitrary length by extending the existing idea of generation through insertion. It received scores of 4466. On one hand, reviewers commented that the paper makes an original contribution by adapting insertion-based sequence generation to general language modeling. The work is of clear presentation and well-structured experiments on both synthetic and real-world datasets. On the other hand, two major concerns regarding efficiency and scaling property still remain. Overall, the AC thinks that the flaws slightly outweigh the merits, and therefore, would like to recommend rejection by the end.

**Reviewer Concerns:**

Concerns adequately addressed:

1. The primary limitation of ILMs is that they sacrifice parallelization benefits from both ARMs and MDMs. Unlike ARMs, ILMs cannot leverage efficient parallel training, and unlike MDMs, they do not support parallel inference. However, despite giving up these parallelization advantages, the improvement in generative perplexity is not substantial (Table 2, LM1B).

2. MDM sampling parameters.

3. Comparison with EditFlow and FlexMDM.

Concerns insufficiently addressed:

1. Efficiency: Any ideas how to address caching/fast inference? The authors have tried to provide a solution with detailed answers, which showed the authors' careful thinking on this question; however, it may not be convincing enough due to a lot of work may be needed to get it work in practice.

2. Scaling: Multiple reviewers showed concerns about the scaling property of the proposed method, such as (1) it's hard to get any signal on this with sequence lengths and complexity as restricted as in e.g. LM1B, and (2) model size scaling. However, the authors' response is less satisfying, and only conducted experiments at 85M, 42M and 8M model size scales.

**Reviewer Scores:**

Overall, I think the authors have provided a detailed rebuttal, but it's still unsure whether by the end the reviewers would be willing to increase the scores due to concerns in scaling and efficiency.

---

### Decision · Program_Chairs · 2026-01-26

Reject